# White-Basilisk: A Hybrid Model for Code Vulnerability Detection

## Abstract

The proliferation of software vulnerabilities presents a significant challenge to cybersecurity, necessitating more effective detection methodologies. We introduce White-Basilisk, a hybrid approach to vulnerability detection that demonstrates strong performance with efficient architectural design. Utilizing an architecture that integrates Mamba layers, linear self-attention, and a Mixture of Experts framework, White-Basilisk achieves state-of-the-art results in vulnerability detection tasks with a parameter count of only 200M. The model's capacity to process extended sequences enables comprehensive analysis of large codebases in a single pass, addressing context limitations that affect current approaches. White-Basilisk exhibits robust performance on imbalanced, real-world datasets, while maintaining computational efficiency that facilitates deployment across diverse organizational scales. This research establishes new benchmarks in code security and provides empirical evidence that compact, efficiently designed models can achieve competitive performance on specialized tasks, contributing to our understanding of architectural efficiency in domain-specific applications.

## 1 Introduction

Recent advancements in artificial intelligence have been largely driven by the rapid expansion of neural networks, where cutting-edge performance is often linked to models that scale to hundreds of billions of parameters. While this scaling paradigm has yielded impressive capabilities, it has also introduced significant computational, environmental, and accessibility challenges (Patterson et al., 2021; Faiz et al., 2023). As the field continues to prioritize ever-larger models, an important question emerges: are there domains where architectural innovation and targeted design might outperform sheer parameter count? This question becomes particularly relevant for high-stakes applications with specific computational constraints, where the "bigger is better" approach may prove suboptimal or economically infeasible.

Code vulnerability detection represents precisely such a domain—one with substantial practical importance and unique technical challenges. The economic implications are significant: cybercrime is projected to reach $10.5 trillion by 2025 (Cybersecurity Ventures, 2023). This massive economic impact is largely preventable, as the Ponemon Institute found that 60% of breach victims were compromised due to an unpatched known vulnerability, while 62% of organizations reported they were completely unaware that their systems were vulnerable prior to a data breach (Ponemon Institute, 2018). When vulnerabilities are found in third-party software, the financial consequences are severe, costing organizations an average of $4.55 million per breach incident (UpGuard, 2023). Despite this growing threat landscape and expanding attack surface resulting from increasingly complex software systems, existing automated vulnerability detection approaches face significant limitations. Traditional static application security testing (SAST) tools demonstrate insufficient detection rates (Zhou et al., 2024), while recent machine learning approaches based on large language models (LLMs) require prohibitive computational resources, making them impractical for many real-world deployment scenarios. Moreover, detecting vulnerabilities often requires analysing extremely long code sequences and capturing subtle interactions spanning multiple functions or files—capabilities that remain challenging even for the largest models due to context length constraints and inefficient attention mechanisms.

Current research has explored various approaches to address these challenges. Transformer-based models fine-tuned on code have shown promising results (Feng et al., 2020; Guo et al., 2022), but their quadratic attention complexity limits their application to large codebases. Specialized vulnerability detection architectures (Hanif & Maffeis, 2022) improve upon these foundations but still struggle with long-range dependencies and resource efficiency. Notably, even as these models grow in size and complexity, their performance on realistic vulnerability detection benchmarks often falls short of practical requirements (Ding et al., 2024). This disconnect between model scale and real-world effectiveness suggests an opportunity to re-examine fundamental architectural assumptions in the context of this specialized task.

We introduce White-Basilisk, a compact 200M-parameter model that challenges conventional scaling wisdom by outperforming substantially larger models on code vulnerability detection tasks. Our approach diverges from the standard practice of scaling existing architectures, instead focusing on designing a specialized hybrid architecture that addresses the unique requirements of vulnerability detection. The key insight driving our work is that by combining state-space models (Mamba) with linear-scaling attention mechanisms and sparsely-activated Mixture of Experts, we can efficiently process extremely long code sequences while capturing both local syntactic patterns and global dependencies crucial for vulnerability identification.

Our contributions can be summarized as follows:

1. **Novel Hybrid Architecture:** We propose a resource-efficient model integrating Mamba layers for local pattern recognition, linear-scaling attention for global context modeling, and conditional computation through Mixture of Experts, enabling effective vulnerability detection with just 200M parameters.

2. **Efficient Context Processing:** White-Basilisk is capable of processing extremely long sequences. It achieves this with a 24-fold reduction in memory consumption when compared to the quadratic attention mechanisms common in standard transformer architectures.

3. **State-of-the-Art Performance:** Through rigorous evaluation across five established benchmarks (PRIMEVUL Ding et al. (2024), BigVul Fan et al. (2020), Draper Russell et al. (2018), REVEAL Chakraborty et al. (2021), and VulDeepecker Li et al. (2018)), we demonstrate that White-Basilisk outperforms models up to 35× larger, particularly on realistic, imbalanced datasets representative of real-world code security scenarios.

Beyond the immediate applications in software security, our work presents broader implications for the field of AI. By demonstrating that a carefully designed domain-specific architecture can outperform general-purpose scaled models, we challenge the notion that parameter count is the primary determinant of model capability. White-Basilisk serves as evidence that the path toward more capable AI systems may lie not only in scaling existing architectures but also in rethinking fundamental design choices for specific high-value applications.

## 2 RELATED WORK

The field of automated code vulnerability detection provides an excellent context for exploring efficient AI architectures. While this domain has undergone rapid evolution, most approaches have followed the general AI trend toward larger models with increasing computational requirements. Our work challenges this paradigm by demonstrating that carefully designed smaller models can achieve superior results. To properly position our contribution, we review both traditional vulnerability detection approaches and recent trends in efficient language modeling.

Driven by the growing complexity and security risks of software systems, early pioneers explored static analysis techniques Chess & McGraw (2004) and pattern matching methods Livshits & Lam (2005). While these approaches laid a foundational framework, they often encountered significant drawbacks, including high false positive rates, difficulty detecting complex vulnerabilities, and susceptibility to obfuscation techniques. Due to these limitations, they were generally unsuitable for active production environments.

As the field matured, researchers recognized the transformative potential of AI, gradually shifting from conventional practices to a new paradigm. This transition was marked by the development of

VulDeePecker Li et al. (2018), one of the first deep learning (DL)-based systems for vulnerability detection. VulDeePecker utilized code gadgets and Bidirectional Long Short-Term Memory (BiLSTM) networks to identify vulnerabilities in C/C++ code. This work demonstrated the ability of DL techniques to capture complex patterns associated with code vulnerabilities, paving the way for the development of further ML-driven solutions. However, its reliance on manually crafted features limited its generalizability. Building on this work, Russell et al. (2018) developed the Draper dataset, which provides a substantial real-world dataset specifically designed for training neural networks in the task of vulnerability detection. Their work showed the advantages of leveraging vast training data to enhance model performance, improving performance but still struggling with limited context windows that restricted the capture of long-range dependencies in code.

Following that, pre-trained LLMs emerged as a significant breakthrough in code analysis. Hanif & Maffeis (2022) proposed VulBERTa, an adaptation of the RoBERTa model for detecting code vulnerabilities, demonstrating the potential of transfer learning from natural language processing to code analysis. By fine-tuning LLMs pre-trained on extensive corpora of code, this approach quickly gained popularity due to its ability to capture latent patterns in both code structure and semantics. However, similar to many transformer-based models, it suffers from quadratic computational complexity with sequence length, constraining its applicability to large-scale projects.

Comprehensive studies by Chakraborty et al. (2021) and Ding et al. (2024) highlighted persistent challenges in the field, including data quality issues, unrealistic evaluation methods, and difficulties in handling long-range dependencies. Ding et al. (2024) showed that existing benchmarks significantly overestimate model performance, with state-of-the-art models achieving high scores on flawed datasets but failing on more realistic ones.

To address the computational and architectural challenges identified in existing vulnerability detection approaches, White-Basilisk's architecture builds upon three established techniques that address complementary challenges in efficient sequence modeling.

State-space models, exemplified by Mamba (Gu & Dao, 2023), offer an alternative to traditional recurrent and attention-based architectures for sequence modeling. Mamba employs a selective state-space mechanism that maintains linear computational complexity with respect to sequence length, contrasting with the quadratic complexity of standard transformer attention. The selective mechanism allows the model to focus on relevant information while efficiently processing long sequences.

Linear attention mechanisms address the computational bottleneck of standard self-attention, which scales quadratically with sequence length. Several approaches have been proposed to achieve linear complexity, including low-rank approximations (Wang et al., 2020) and kernel-based methods. Infini-attention (Munkhdalai et al., 2024) represents a recent advancement in this direction, introducing a memory-based approach that maintains both local attention patterns and compressed representations of distant context. These mechanisms aim to preserve the modeling capabilities of attention while reducing computational requirements, though the effectiveness varies depending on the specific task and sequence characteristics.

Mixture of Experts (MoE) architectures introduce conditional computation by activating only a subset of model parameters for each input token (Fedus et al., 2022). This approach allows models to increase their capacity without proportionally increasing computational cost during inference. Recent implementations like Jamba (Lieber et al., 2024) have demonstrated the effectiveness of combining MoE with both attention and state-space components in a hybrid architecture. The sparse activation pattern in MoE can provide computational efficiency while maintaining model expressiveness.

## 3 MODEL ARCHITECTURE

White-Basilisk addresses a fundamental limitation in current language models: the quadratic complexity of standard attention mechanisms. Our architecture introduces a hybrid approach that achieves linear complexity with respect to sequence length while maintaining the representational power necessary for complex reasoning tasks. The core innovation lies in combining three complementary components: Mamba layers, a novel linear-scaling attention mechanism, and Mixture of Experts. This combination enables processing long sequences with substantially reduced memory requirements compared to standard transformers.

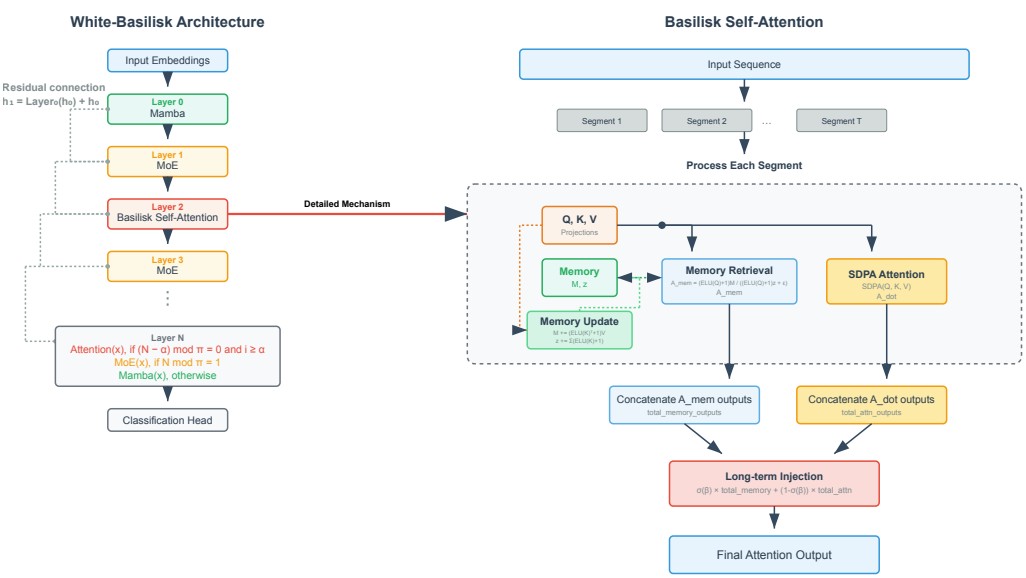

Figure 1: White-Basilisk Model architecture

### 3.1 BASILISK SELF-ATTENTION

The most significant architectural innovation in White-Basilisk is our linear-scaling attention mechanism, which we term Basilisk Self-Attention. Our approach builds upon recent advances in linear attention, particularly the Infini-attention framework (Munkhdalai et al., 2024), but introduces fundamental modifications that better suit general-purpose language modeling. While the original Infini-attention processes sequences in independent segments, our adaptation maintains global context through cumulative information aggregation across the entire sequence.

The mechanism operates by dividing long input sequences into segments of fixed size $S$ (typically 2,048 tokens), a segment size validated by (Munkhdalai et al., 2024). For each segment $s$, we compute both local attention patterns and retrieve information from a compressed global memory. The key insight is that rather than processing segments independently, we accumulate context information across the entire sequence, enabling attention to span arbitrarily long distances while maintaining linear complexity with respect to sequence length.

Mathematically, our attention mechanism can be expressed through the following operations. For a sequence divided into $T$ segments, we maintain running totals of memory-based and attention-based components:

$$\text{total}_{\text{mem}} = \sum_{s=1}^{T} A_{\text{mem},s}, \quad \text{total}_{\text{attn}} = \sum_{s=1}^{T} A_{\text{dot},s} \tag{1}$$

where $A_{\text{mem},s}$ represents the memory-based attention output for segment $s$, and $A_{\text{dot},s}$ represents the standard dot-product attention output for segment $s$. This accumulation process ensures that information from all previous segments remains accessible to current computations.

The final output combines these accumulated components through an adaptive gating mechanism:

$$O = \text{sigmoid}(\beta) \odot \text{total}_{\text{mem}} + (1 - \text{sigmoid}(\beta)) \odot \text{total}_{\text{attn}} \tag{2}$$

where $\beta$ is a learnable parameter that determines the balance between memory-based and direct attention components, and $\odot$ denotes element-wise multiplication.

The memory-based component implements a compressed representation of key-value information from previously processed segments. For each segment, we compute:

$$A_{\text{mem}} = \frac{(\text{ELU}(Q) + 1)M^T}{(\text{ELU}(Q) + 1)z^T + \epsilon} \tag{3}$$

where $Q$ represents the query matrix for the current segment, $M$ is the compressive memory matrix, $z$ is a normalization term, $\epsilon$ is a small constant for numerical stability, and ELU is the Exponential Linear Unit activation function. The ELU activation ensures non-negative values, which is crucial for the probabilistic interpretation of attention weights.

The memory matrix $M$ and normalization term $z$ are updated incrementally as each segment is processed:

$$M_{\text{new}} = M + (\text{ELU}(K)^T + 1)V, \quad z_{\text{new}} = z + \sum_{i=1}^{S}(\text{ELU}(K_i) + 1) \tag{4}$$

where $K$ and $V$ are the key and value matrices for the current segment, and $S$ is the segment length.

This approach differs fundamentally from other linear attention methods such as Linformer (Wang et al., 2020), which reduces attention complexity through low-rank approximations, or Performer (Choromanski et al., 2022), which uses random feature maps. While these methods achieve linear complexity, they typically sacrifice some representational capacity. Our memory-based approach maintains rich contextual information through the compressed memory mechanism while achieving $O(nSd + 2nd^2 + 2nd)$ computational complexity. The original Infini-attention maintains constant $O(d^2)$ memory footprint by processing segments in streaming fashion, our approach accumulates segment outputs before final concatenation, resulting in $O(nd) + O(d^2)$ memory complexity. This design choice trades the bounded memory property for simplified implementation and improved global context integration, though it limits processing to sequences that fit within available memory rather than enabling truly infinite sequence processing. Detailed computational and memory complexity analysis is provided in Appendix F.

### 3.2 INTEGRATION WITH MAMBA AND MIXTURE OF EXPERTS

White-Basilisk employs a structured interleaving of three layer types: Mamba layers, our linear-scaling attention layers, and Mixture of Experts (MoE) layers. Mamba layers provide the backbone for local pattern recognition and efficient sequence processing. These layers implement selective state-space modeling with linear computational complexity, making them well-suited for capturing local dependencies in long sequences. The MoE layers introduce adaptive computation by activating only a subset of parameters for each token. This design reduces computational cost compared to dense layers of equivalent capacity while maintaining model expressiveness.

The layer combination follows a systematic pattern inspired by Lieber et al. (2024) and defined by two parameters: attention layer offset ($\alpha = 2$) and attention layer period ($\pi = 8$), with these specific parameter values validated by Lieber et al. (2024). The layer type for position $i$ is determined by:

$$\text{Layer}_i = \begin{cases} \text{Attention}(x), & \text{if } (i - \alpha) \bmod \pi = 0 \text{ and } i \geq \alpha \\ \text{MoE}(x), & \text{if } i \bmod a = 1 \\ \text{Mamba}(x), & \text{otherwise} \end{cases} \tag{5}$$

This interleaving pattern ensures regular insertion of global attention layers while maintaining efficient local processing through Mamba layers. The MoE layers are interspersed to provide adaptive computation where beneficial.

The forward pass through White-Basilisk can be described as a sequence of transformations applied to the hidden state. Let $h_i$ denote the hidden state after layer $i$, where $h_0$ represents the input embeddings. Each layer applies its transformation while preserving information flow through residual connections:

$$h_i = \text{Layer}_i(h_{i-1}) + h_{i-1} \tag{6}$$

The residual connections serve two crucial purposes: they facilitate gradient flow during training, preventing the vanishing gradient problem in deep networks, and they preserve information across layers, allowing the model to selectively refine representations without losing previously computed features.

The resulting architecture maintains linear complexity with respect to sequence length while providing the representational capacity necessary for complex reasoning tasks. This design enables processing of sequences that would be computationally infeasible with standard transformer architectures, opening new possibilities for applications requiring extensive context understanding.

## 4 PRETRAINING

Traditional LLM training methods are generally designed to enable models to comprehend language and its syntax. This is often accomplished through Causal Language Modeling (CLM), where a model learns to predict the next token given its input. Another common approach is based on the Fill in the Middle (FIM) technique, in which random text portions are masked, and the model must reconstruct the missing content. Some advanced source code LLMs combine both methods to increase model flexibility (Li et al., 2023). Similarly, in our work, we employ both techniques during model pre-training on the selected 2M code samples. This process requires approximately 600 hours to complete on 1x NVIDIA A100 40GB GPU and emitted approximately 85 kg of $CO_2$.

For pretraining data, we initially selected a carefully curated subset of the StarCoder dataset (Li et al., 2023), which includes more than 80 programming languages and consists of 305M files in total. For our study, we focused on C and C++ code samples, using 2M code samples during the pre-training phase. **Crucially, our pretraining stage used exclusively general-purpose code from the StarCoder dataset with no exposure to any vulnerability detection datasets or vulnerability-specific annotations.** This separation ensures that our model's vulnerability detection capabilities emerge solely from the fine-tuning stage rather than from pretraining exposure to security-related patterns.

## 5 FINE-TUNING AND EVALUATION

To evaluate the pre-trained model, we required well-established benchmarking datasets with publicly available partitions for fine-tuning and testing. This ensures a fair comparison with existing methods without the need to recreate the original models. For this purpose, we selected five publicly available datasets: VulDeePecker (Li et al., 2018), Draper (Russell et al., 2018), PrimeVul (Ding et al., 2024), REVEAL (Chakraborty et al., 2021), and BigVul (Fan et al., 2020).

These datasets present significant challenges that validate White-Basilisk's capabilities. They feature significant class imbalance with the vulnerable class representing only 2.9-9.8% of samples, and diverse vulnerability types covering multiple Common Weakness Enumeration (CWE) categories.

**Our fine-tuning methodology follows a rigorous evaluation protocol to prevent overfitting to evaluation characteristics.** For each dataset, we fine-tuned our pretrained model on the binary classification task (vulnerable vs. non-vulnerable code) using only the publicly available training splits. During training, we monitored performance on the test splits to select the best-performing checkpoint based on F1 score. Finally, we evaluated the selected checkpoint on the validation splits to obtain our reported results. This approach ensures that our final performance metrics represent genuine held-out evaluation without contamination from validation set characteristics.

To address the substantial class imbalance present in vulnerability detection datasets, we implemented three key techniques and selected appropriate evaluation metrics. For training, we applied class weighting using inverse frequency weighting to balance loss contributions between vulnerable and non-vulnerable samples, employed a Weighted Random Sampler without replacement to ensure balanced representation within each training batch, and incorporated Scale-Invariant Fine-Tuning (SIFT) to improve model robustness against small perturbations. For evaluation, we prioritized metrics that provide balanced assessment of model performance, using F1 Score as our primary metric to balance precision and recall, and the Vulnerability Detection Score (VD-S) introduced by (Ding et al., 2024), which measures False Negative Rate. Detailed descriptions of these techniques are provided in Appendix B.5.

## 6 EXPERIMENTAL RESULTS

To comprehensively evaluate White-Basilisk's effectiveness, we conducted experiments across multiple dimensions: dataset-specific performance, cross-dataset generalization, paired vulnerability detection, context length analysis, and vulnerability type classification.

### 6.1 MAIN RESULTS: DATASET-SPECIFIC AND UNIFIED TRAINING

We evaluated White-Basilisk using two training strategies: dataset-specific models (trained and evaluated on individual datasets **independently**) and a unified model (trained on combined train data spits from all datasets and evaluated on the validation set of each dataset).

Table 1: Performance comparison on PRIMEVUL and BigVul datasets

| Model | PRIMEVUL | | | BigVul | | |
|---|---|---|---|---|---|---|
| | Acc | F1 | VD-S ↓ | Acc | F1 | VD-S ↓ |
| CodeT5 | 0.967 | 0.197 | 0.899 | 0.957 | 0.649 | 0.773 |
| CodeBERT | 0.969 | 0.209 | 0.888 | 0.956 | 0.629 | 0.818 |
| UnixCoder | 0.969 | 0.214 | 0.892 | 0.965 | 0.655 | 0.623 |
| StarCoder2 | 0.970 | 0.181 | 0.896 | 0.962 | 0.683 | 0.691 |
| CodeGen2.5 | 0.967 | 0.196 | 0.915 | 0.966 | 0.673 | 0.617 |
| White-Basilisk (Dataset-Specific) | **0.963** | **0.291** | **0.724** | **0.994** | **0.949** | **0.040** |
| White-Basilisk (Unified) | 0.952 | 0.233 | 0.767 | 0.993 | 0.938 | 0.062 |

Table 2: Performance on traditional vulnerability detection benchmarks

| Model | Draper | REVEAL | | VulDeePecker | |
|---|---|---|---|---|---|
| | F1 | Acc | F1 | F1 | Precision |
| Russell et al. (2018) | 0.566 | - | - | - | - |
| VulBERTa-MLP | 0.433 | 0.845 | 0.453 | 0.930 | 0.958 |
| VulBERTa-CNN | 0.579 | 0.797 | 0.426 | 0.909 | 0.953 |
| Baseline-BiLSTM | 0.468 | 0.771 | 0.391 | 0.670 | 0.526 |
| Baseline-TextCNN | 0.494 | 0.732 | 0.374 | 0.758 | 0.635 |
| REVEAL | - | 0.844 | 0.413 | - | - |
| VulDeePecker | - | - | - | 0.929 | 0.919 |
| White-Basilisk (Dataset-Specific) | **0.607** | **0.899** | **0.493** | **0.939** | **0.972** |
| White-Basilisk (Unified) | 0.549 | 0.888 | 0.470 | 0.925 | 0.939 |

White-Basilisk consistently outperforms all baseline models across every vulnerability detection benchmark. On PRIMEVUL, our dataset-specific model achieves F1=0.291, representing a 35.5% improvement over the best baseline (UnixCoder: F1=0.214). On BigVul, White-Basilisk achieves F1=0.949, dramatically outperforming all competing approaches. The model demonstrates similar superiority on traditional benchmarks.

The unified model demonstrates remarkable cross-dataset generalization capability, maintaining competitive performance across all benchmarks despite being trained on combined data from diverse sources. Performance differences between dataset-specific and unified training are modest, suggesting effective knowledge transfer without significant performance degradation. This generalization spans datasets with different characteristics demonstrating the robustness of our approach.

Analysis of vulnerability types reveals that White-Basilisk demonstrates strong performance across different Common Weakness Enumeration (CWE) categories, with particularly robust detection of memory-related vulnerabilities including CWE-119 (buffer overflows) and CWE-476 (NULL pointer dereferences). The unified model achieves perfect F1=1.000 scores for 22 different CWE categories in BigVul evaluation, with consistently high performance on access control issues (CWE-264: F1=0.925) and input validation vulnerabilities (CWE-20: F1=0.933). Detailed CWE-specific performance analysis is provided in Appendix I.

## 6.2 PRIMEVUL PAIRED EVALUATION

To address concerns about potential dataset artifacts and superficial pattern recognition, we evaluated White-Basilisk on the PRIMEVUL paired dataset, which consists of vulnerable code paired with its patched counterpart. This evaluation specifically tests the model's ability to distinguish between vulnerable code and its security fix, often involving subtle changes that challenge pattern-matching approaches. The evaluation uses four metrics: P-C measures correct classification of both vulnerable and patched code in each pair, P-V indicates cases where both functions are incorrectly classified as vulnerable, P-B represents cases where both functions are incorrectly classified as benign, and P-R captures complete prediction reversals where vulnerable code is classified as safe and patched code as vulnerable.

Table 3: PRIMEVUL paired evaluation results

| Model | Method | P-C ↑ | P-V ↓ | P-B ↓ | P-R ↓ |
|---|---|---|---|---|---|
| White-Basilisk (Ours) | Fine-tuning | **12.92** | 42.08 | 42.92 | 2.08 |
| GPT-4 | Chain-of-Thought | 12.94 | 54.26 | 24.47 | 8.33 |
| GPT-4 | Two-shot | 5.14 | 71.63 | 21.45 | 1.77 |
| CodeGen2.5 | Fine-tuning | 3.01 | 10.82 | 84.22 | 1.95 |
| StarCoder2-7B | Fine-tuning | 2.30 | 8.16 | 88.30 | 1.24 |
| CodeBERT | Fine-tuning | 1.77 | 11.35 | 86.17 | 0.71 |
| Random Baseline | - | 22.70 | 26.24 | 26.42 | 24.65 |

White-Basilisk achieves 12.92% P-C performance, representing a 4.29× improvement over the best open-source baseline (CodeGen2.5: 3.01%) and nearly matching GPT-4 with chain-of-thought reasoning (12.94%). The balanced error distribution between P-V (42.08%) and P-B (42.92%) indicates active vulnerability reasoning rather than biased default predictions. However, all models perform substantially below random baseline (22.70%), highlighting the fundamental difficulty of this task.

## 6.3 CONTEXT LENGTH IMPACT

We conducted two experiments to validate our architectural choices and context processing claims. First, we evaluated our unified model across context lengths from 512 to 131K tokens on BigVul and PRIMEVUL datasets. Second, we performed full-file analysis on PRIMEVUL by cross-matching validation samples with their complete C/C++ source files and classifying entire files (up to 524K tokens) rather than isolated vulnerable functions.

Table 4: Performance across different context lengths and full-file validation

| Context Length | Baseline Models Context | BigVul | | PRIMEVUL | |
|---|---|---|---|---|---|
| | | Acc | F1 | Acc | F1 |
| 512 | CodeBERT, CodeT5 | 0.974 | 0.752 | 0.958 | 0.142 |
| 1,024 | UnixCoder, VulBERTa | 0.984 | 0.855 | 0.954 | 0.156 |
| 2,048 | CodeGen2.5 | 0.990 | 0.909 | 0.953 | 0.205 |
| 4,096 | StarCoder2 (sliding) | 0.992 | 0.933 | 0.953 | 0.219 |
| 16,384 | StarCoder2 (max) | 0.993 | 0.942 | 0.952 | 0.235 |
| 32,768 - 131,072 | White-Basilisk | 0.993 | 0.943 | 0.952 | 0.233 |
| **Full-File Context Validation** | | | | | |
| 524,288 | PRIMEVUL Full Files | - | - | **0.956** | **0.288** |

The context length experiment shows consistent performance improvements from 512 to 16K tokens (BigVul F1: 0.752→0.942; PRIMEVUL F1: 0.142→0.235), after which performance plateaus. This pattern aligns with our dataset analysis showing most samples fall within the 16K token range. The full-file experiment demonstrates a substantial F1 improvement (0.233→0.288) when analyzing complete source files instead of isolated functions.

These results validate two key aspects of our architecture. The linear scaling with context length confirms that vulnerabilities benefit from broader contextual information, while the dramatic improvement in full-file analysis indicates that many vulnerabilities manifest through inter-function

dependencies invisible in isolated code fragments. The ability to process 524K-token sequences demonstrates the practical utility of our linear-scaling attention mechanism for complete codebase analysis, a capability that would be computationally infeasible with standard quadratic attention approaches. Additional sequence length analysis is provided in Appendix H.2.

## 7 LIMITATIONS AND FUTURE WORK

While White-Basilisk shows promising results, several limitations constrain its current applicability. The primary limitation is its focus on C and C++ codebases, with unknown generalization to other programming languages with different syntaxes and vulnerability patterns. Combined with potential dataset biases, this may limit effectiveness on diverse real-world codebases.

Two fundamental dataset limitations constrain our evaluation. Current benchmarks contain primarily function-level code snippets rather than file-level samples, potentially missing crucial contextual information where vulnerabilities emerge from interactions between multiple functions or modules. Many vulnerabilities appear benign in isolation but become dangerous through complex inter-function dependencies. Additionally, existing datasets contain mostly sequences under 16K tokens, preventing full demonstration of White-Basilisk's extended context capability and long-range dependency modeling. While our ablation studies show consistent improvements from 512 to 16K tokens, the scarcity of longer sequences means we cannot fully validate our architecture's full potential. These represent broader field limitations rather than specific shortcomings of our approach.

Despite strong benchmark performance, White-Basilisk faces challenges common to automated vulnerability detection. False positives reduce developer trust while false negatives leave vulnerabilities undetected. Performance on novel vulnerability types or zero-day exploits remains uncertain. The model's explainability is limited, it provides binary classifications without indicating reasoning, vulnerability locations, or risk assessments, creating barriers for security-critical deployment where developers need actionable insights.

We plan to address dataset limitations by developing comprehensive benchmarks with file-level samples preserving contextual information and longer sequences exercising our extended context capabilities. However, we first needed to establish performance on existing benchmarks for fair comparison with prior work. Future research will enhance explainability through attention visualization and program analysis integration, scale to approximately 1 billion parameters while maintaining efficiency, and investigate our hybrid architecture's applicability to broader NLP tasks requiring efficient long-sequence processing.

## 8 CONCLUSION

White-Basilisk introduces a hybrid architecture that integrates Mamba layers, linear-scaling attention, and Mixture of Experts to achieve linear computational scaling with sequence length. This architectural design enables processing of extended sequences while maintaining compact parameter count, demonstrating that efficient architectural choices can overcome the quadratic complexity limitations of standard transformers.

The model achieves state-of-the-art performance across five vulnerability detection benchmarks, consistently outperforming substantially larger models. The significant performance gains from full-file context analysis validate that vulnerability detection benefits substantially from extended context processing capabilities that span multiple functions and broader code structure. These results establish that linear-complexity architectures can deliver superior performance on challenging downstream tasks requiring long-range dependencies. Code vulnerability detection represents a particularly demanding application where subtle patterns must be identified across extensive sequences, making it an effective testbed for evaluating architectural efficiency versus capability trade-offs.

The hybrid approach demonstrates broader applicability beyond security applications. Tasks requiring analysis of long documents, complex system modeling, or extended contextual understanding can benefit from similar architectural principles that maintain linear complexity with respect to sequence length while preserving representational capacity. The compact design enables deployment in resource-constrained environments without sacrificing performance, potentially expanding access to advanced sequence processing capabilities across diverse applications and computational budgets.

## 9 REPRODUCIBILITY STATEMENT

To ensure reproducibility of our results, we provide comprehensive implementation details through-out this paper and supplementary materials. Section 3 presents the complete White-Basilisk ar-chitecture specification, including our novel Basilisk Self-Attention mechanism with mathematical formulations (Equations 1-7). Appendix B contains all hyperparameters used for both pretraining and fine-tuning phases, including model dimensions, learning rates, batch sizes, and optimization settings. The pretraining methodology using 2M C/C++ samples from the StarCoder dataset is de-tailed in Section 4, while Section 5 describes our fine-tuning protocol and evaluation methodology across five public benchmarks. Appendix B.1 provides the complete classification head architecture, and Appendix B.5 details our class imbalance handling techniques including SIFT implementation. Our computational complexity analysis in Appendix F includes empirical memory consumption measurements to validate theoretical claims. All datasets used (PRIMEVUL, BigVul, Draper, RE-VEAL, VulDeepecker) are publicly available with the exact train/validation/test splits referenced in our evaluation. The training infrastructure (single NVIDIA A100 40GB GPU, 600 training hours, 85 kg CO2 emissions) and complete experimental setup are documented to facilitate replication of our 200M-parameter model.

To further support reproducibility and encourage additional research using our architecture, we will publicly release the complete White-Basilisk model weights, training scripts, inference code, and preprocessing pipelines upon paper acceptance. This release will include comprehensive documen-tation, example usage scripts, and step-by-step instructions for both reproducing our experimental results and adapting the architecture for new applications. Additionally, we will provide configu-ration files for all experimental settings, evaluation scripts for the benchmark datasets, and detailed environment specifications to ensure seamless replication across different computational environ-ments.

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

# A APPENDIX

# B ARCHITECTURAL DETAILS AND HYPERPARAMETERS

For reproducibility purposes we provide all the model parameters and training hyperparameters in this section:

## B.1 CLASSIFICATION HEAD

The classification head of White-Basilisk is designed to efficiently transform the high-dimensional representations learned by the main model into classification outputs for vulnerability detection. Its architecture is as follows:

1. **Dense Layer 1:** A fully connected layer that projects the hidden state (dimension 512) to the same dimension. This layer uses a GELU activation function and is followed by dropout for regularization.

2. **Dense Layer 2:** Another fully connected layer that reduces the dimension from 512 to 256, again followed by GELU activation and dropout.

3. **Layer Normalization:** Applied to the output of Dense Layer 2 for improved stability and faster convergence.

4. **Output Layer:** A final linear layer that projects from 256 dimensions to the number of classes (typically 2 for binary classification of vulnerable vs. non-vulnerable code).

This classification head structure was chosen to gradually reduce the dimensionality of the representations while maintaining the model's ability to capture complex patterns relevant to vulnerability detection. The use of GELU activations and layer normalization aligns with modern best practices in deep learning architecture design. The classification head is mathematically described as follows:

$$
\begin{aligned}
x_1 &= \text{Dropout}(\text{GELU}(W_1 h + b_1)) \\
x_2 &= \text{Dropout}(\text{GELU}(W_2 x_1 + b_2)) \\
x_3 &= \text{LayerNorm}(x_2) \\
y &= W_3 x_3 + b_3
\end{aligned}
\tag{7}
$$

where $h \in \mathbb{R}^{512}$ is the input from the main model, $W_1 \in \mathbb{R}^{512 \times 512}$, $W_2 \in \mathbb{R}^{256 \times 512}$, and $W_3 \in \mathbb{R}^{2 \times 256}$ are learnable weights, and $b_1, b_2, b_3$ are biases.

## B.2 HYPERPARAMETER DETAILS

This section provides a detailed overview of the hyperparameters used in training White-Basilisk, including both the pretraining and fine-tuning phases. We also discuss the rationale behind key hyperparameter choices and their impact on model performance.

## B.3 PRETRAINING HYPERPARAMETERS

**Learning Rate:** We chose a relatively small learning rate of 1.41e-5 to ensure stable training given the complexity of the task and the hybrid nature of our model architecture. This value was determined through careful tuning to balance training speed and convergence stability.

**Batch Size:** A batch size of 16 was selected as a compromise between training efficiency and memory constraints of our hardware (single NVIDIA A100 40GB GPU). Larger batch sizes could potentially improve training stability but would require more memory or gradient accumulation steps.

**Number of Epochs and Warmup Ratio:** We trained for 10 epochs with a warmup ratio of 0.15. This combination allowed the model to reach good performance while preventing overfitting. The warmup period helps stabilize training in the early stages.

**Optimizer Settings:** We used the AdamW optimizer with $\beta_1 = 0.9$, $\beta_2 = 0.999$, and $\epsilon = 1e - 8$. These are standard settings that work well across a wide range of tasks. The weight decay of 0.01 was applied to all parameters except for bias and LayerNorm weights to prevent overfitting.

Table 5: Hyperparameters for the White-Basilisk Model

| Hyperparameter | Value |
|---|---|
| Vocabulary Size | 65,540 |
| Hidden Size | 512 |
| Intermediate Size | 1,792 |
| Number of Hidden Layers | 12 |
| Hidden Activation | SiLU |
| Initializer Range | 0.02 |
| RMS Norm Epsilon | $1 \times 10^{-6}$ |
| **Attention Parameters** | |
| Attention Layer Offset | 2 |
| Attention Layer Period | 8 |
| Number of Attention Heads | 16 |
| Number of Key-Value Heads | 8 |
| **Mixture of Experts (MoE) Parameters** | |
| Expert Layer Offset | 1 |
| Expert Layer Period | 2 |
| Number of Experts | 8 |
| Experts per Token | 2 |
| Router Auxiliary Loss Coefficient | 0.001 |
| **Mamba Parameters** | |
| Mamba Convolution Dimension | 4 |
| Mamba State Dimension | 16 |
| Mamba Delta Rank | 256 |
| Mamba Expansion Factor | 2 |

**FIM and FIM-SPM Rates:** Both the Fill-in-the-Middle (FIM) rate and the FIM Sentence Permutation Mode (SPM) rate were set to 0.5. This means that 50% of the samples undergo FIM transformation, and among those, 50% use the SPM variant. These rates provide a good balance between standard causal language modeling and the FIM objective, enhancing the model's bidirectional understanding capabilities.

### B.4 FINE-TUNING HYPERPARAMETERS

**Learning Rate and Batch Size:** We used a smaller learning rate (5e-6) and batch size (4) for fine-tuning to prevent catastrophic forgetting and to allow the model to adapt to the specific characteristics of each dataset without overfitting.

**SIFT Parameters:** For Scale-Invariant Fine-Tuning, we used a learning rate of 1e-4 for the perturbation layer and an initial perturbation magnitude of 1e-2. These values were chosen to provide meaningful adversarial examples without overly distorting the input embeddings.

### B.5 HANDLING CLASS IMBALANCE

$$w_c = \frac{N}{2N_c} \tag{8}$$

### B.5.1 WEIGHTED RANDOM SAMPLING IMPLEMENTATION

We implement weighted random sampling using PyTorch's WeightedRandomSampler **WITHOUT** replacement, which samples each example exactly once per epoch but reorders them based on importance weights. This approach provides two key benefit":

1. **Temporal Concentration:** Vulnerable samples appear earlier in the training epoch when gradients are typically more effective for learning

2. **Batch Distribution Optimization:** Maximizes the number of batches that contain vulnerable samples by preventing them from being wastefully clustered together in the same batches

To illustrate the effectiveness of this approach, consider the PRIMEVUL training dataset with 184,427 samples containing 3.02% vulnerable samples (5,569 vulnerable samples) and batch size 4:

- **Total batches per epoch:** 46,107
- **Batches with at least vulnerable sample:** 7.78% (3,585 batches)
- **Batches with 0 vulnerable samples:** 92.22% (42,521 batches)

The weighted sampling with $w_c = \frac{N}{2N_c}$ gives vulnerable samples higher selection probability, ensuring maximum vulnerable-containing batches while front-loading their appearance in the epoch. While severe class imbalance means many batches will still contain only non-vulnerable samples, the weighted sampling optimizes the distribution of available vulnerable samples across the training process.

### B.5.2 WEIGHTED LOSS FUNCTION

We implement a weighted loss function, modifying the standard cross-entropy loss by incorporating the class-specific weights $w_c$. This ensures that the loss contribution from vulnerable samples is appropriately scaled relative to their frequency in the dataset.

### B.5.3 SIFT (SCALE-INVARIANT FINE-TUNING)

We implement automated adversarial training using SIFT to improve the model's resilience. SIFT operates by introducing small perturbations to the input during training, encouraging the model to learn more robust features. In our implementation, we added a PerturbationLayer into the model architecture, which applies learnable perturbations to the input embeddings. The training process was designed to minimize both the task loss and the adversarial loss, the latter being computed as the difference between predictions on clean and perturbed inputs.

This approach confers several advantages, including improved model generalization and enhanced robustness to minor variations in input. Such resilience is crucial in the domain of code vulnerability detection, where the model must maintain consistent performance across diverse code samples and potential adversarial inputs.

## C EVALUATION METRICS

This appendix provides detailed descriptions of all metrics used to evaluate model performance in vulnerability detection tasks.

### C.1 STANDARD CLASSIFICATION METRICS

**Accuracy** Accuracy measures the proportion of correct predictions (both true positives and true negatives) among all predictions:

$$\text{Accuracy} = \frac{TP + TN}{TP + TN + FP + FN} \tag{9}$$

where TP = True Positives, TN = True Negatives, FP = False Positives, and FN = False Negatives.

**Precision** Precision measures the proportion of correct positive predictions among all positive predictions:

$$\text{Precision} = \frac{TP}{TP + FP} \tag{10}$$

**Recall** Recall (also known as sensitivity) measures the proportion of actual positives correctly identified:

$$\text{Recall} = \frac{TP}{TP + FN} \tag{11}$$

**F1 Score**   F1 Score is the harmonic mean of precision and recall, providing a balanced measure of model performance:

$$\text{F1} = 2 \times \frac{\text{Precision} \times \text{Recall}}{\text{Precision} + \text{Recall}} \tag{12}$$

**Vulnerability Detection Score (VD-S)**   VD-S evaluates the False Negative Rate of a detector at a specific False Positive Rate (FPR) threshold:

$$\text{VD-S} = \frac{FN}{FN + TP} \text{ at } FPR \leq 0.005 \tag{13}$$

where a lower score indicates better performance. This metric is particularly important for security applications as it measures the model's ability to minimize missed vulnerabilities while maintaining a low false positive rate.

### C.2   PAIRWISE EVALUATION METRICS

The pairwise evaluation metrics are specifically designed for vulnerability detection tasks involving pairs of vulnerable code and their corresponding patched versions. These metrics assess a model's ability to distinguish between vulnerable code and its security fix, addressing concerns about superficial pattern recognition.

**P-C (Pair-Correct)**   P-C measures the percentage of code pairs where both the vulnerable and patched versions are correctly classified:

$$\text{P-C} = \frac{\text{Pairs with both correct classifications}}{\text{Total pairs}} \times 100\% \tag{14}$$

A higher P-C score indicates better overall discrimination capability between vulnerable and safe code.

**P-V (Pair-Vulnerable)**   P-V indicates the percentage of pairs where both vulnerable and patched code are incorrectly classified as vulnerable:

$$\text{P-V} = \frac{\text{Pairs where both classified as vulnerable}}{\text{Total pairs}} \times 100\% \tag{15}$$

This metric reveals tendency toward false positive predictions and failure to recognize security patches.

**P-B (Pair-Benign)**   P-B represents the percentage of pairs where both vulnerable and patched code are incorrectly classified as benign:

$$\text{P-B} = \frac{\text{Pairs where both classified as benign}}{\text{Total pairs}} \times 100\% \tag{16}$$

This metric indicates conservative prediction behavior and potential missed vulnerabilities.

**P-R (Pair-Reversed)**   P-R captures the percentage of complete prediction reversals where vulnerable code is classified as safe and patched code is classified as vulnerable:

$$\text{P-R} = \frac{\text{Pairs with reversed classifications}}{\text{Total pairs}} \times 100\% \tag{17}$$

This metric identifies the most problematic prediction pattern, suggesting fundamental misunderstanding of vulnerability characteristics.

**Interpretation Guidelines**   For pairwise metrics, the ideal performance would show high P-C values with low P-V, P-B, and P-R values. The distribution of error types (P-V vs. P-B vs. P-R) provides insights into model behavior: balanced P-V and P-B with low P-R suggests active reasoning rather than biased default predictions, while high P-R indicates serious model confusion about vulnerability patterns.

Each metric serves a specific purpose in evaluating different aspects of model performance, from general classification accuracy to specialized vulnerability detection capabilities and robustness against dataset artifacts. The combination of these metrics provides a comprehensive assessment of a model's effectiveness in real-world security applications.

# D  DATASET STATISTICS ANALYSIS

This section provides a comprehensive analysis of five vulnerability detection datasets (PRIMEVUL, BigVul, REVEAL, Draper, and VulDeepecker), examining their size distributions, class imbalance characteristics, and data quality metrics.

Table 6: Dataset Distribution and Vulnerability Statistics

| Dataset | Sample Count | | | Vulnerable (%) | | | Duplicates (%) | | |
|---|---|---|---|---|---|---|---|---|---|
| | Train | Val | Test | Train | Val | Test | Train | Val | Test |
| Draper | 1,019,471 | 127,476 | 127,419 | 6.46 | 6.47 | 6.48 | 0.00 | 0.00 | 0.00 |
| PRIMEVUL | 184,427 | 25,430 | 25,911 | 3.02 | 2.75 | 2.68 | 0.00 | 0.00 | 0.00 |
| BigVul | 150,908 | 18,864 | 18,864 | 5.79 | 5.88 | 5.59 | 0.005 | 0.00 | 0.00 |
| VulDeepecker | 128,118 | 16,015 | 16,015 | 6.08 | 6.08 | 6.08 | 37.72 | 20.27 | 20.33 |
| REVEAL | 18,187 | 2,273 | 2,274 | 9.90 | 9.24 | 10.11 | 1.17 | 0.18 | 0.09 |

Table 7: Sequence Length Statistics (Training Split)

| Dataset | Min | Max | Mean | Median | 95th % |
|---|---|---|---|---|---|
| PRIMEVUL | 3 | 296,924 | 502.44 | 193.0 | 1,729.0 |
| VulDeepecker | 8 | 312,940 | 284.03 | 142.0 | 893.0 |
| REVEAL | 10 | 120,684 | 569.05 | 226.0 | 1,929.7 |
| BigVul | 6 | 70,440 | 343.90 | 147.0 | 1,193.0 |
| Draper | 10 | 42,492 | 320.85 | 236.0 | 858.0 |

## D.1  IMPLICATIONS FOR MODEL DESIGN

These statistics significantly influenced our model design decisions:

1. The substantial class imbalance across all datasets (ranging from 3.02% to 10.11% vulnerable samples) motivated our implementation of specialized class weighting and sampling strategies.

2. The extreme range in sequence lengths (from 3 to 312,940 tokens) justified our focus on developing an architecture capable of handling very long sequences efficiently.

3. The varying levels of data duplication (0% to 37.72%) highlighted the importance of robust evaluation metrics and careful interpretation of results, particularly for VulDeepecker.

4. The consistency of class distributions across splits suggests that our evaluation metrics should be reliable indicators of real-world performance.

5. REVEAL's higher proportion of vulnerable samples (10%) compared to other datasets (3-6%) provides an important test case for our model's ability to handle different class balance scenarios.

# E  BASELINE MODELS

This section details the baseline models examined in our study. It is important to note that we did not train, finetune, or run any of these models ourselves. Instead, we collected and analyzed their reported metrics and configurations from their respective papers. For CodeT5 (CT5), CodeBERT (CB), UnixCoder (UC), StarCoder2 (SC2), and CodeGen2.5 (CG2.5), the information was sourced from the PrimeVul paper Ding et al. (2024). For VulBERTa variants (VulBERTa-MLP and VulBERTa-CNN) and the baseline models (Baseline-BiLSTM and Baseline-TextCNN), the information was obtained from the original VulBERTa paper Hanif & Maffeis (2022).

Table 8: Overview of baseline models examined in our study

| Model | Architecture | Pre-training | Params |
|---|---|---|---|
| CT5 Wang et al. (2021) | Enc-Dec | Multi-lingual code | 220M |
| CB Feng et al. (2020) | Encoder | Bimodal (code + text) | 125M |
| UC Guo et al. (2022) | Encoder | Cross-modal | 125M |
| SC2 Li et al. (2023) | Decoder | The Stack v2 | 7B |
| CG2.5 Nijkamp et al. (2023) | Decoder | Code + natural lang. | 7B |
| VulBERTa-MLP Hanif & Maffeis (2022) | Encoder | C/C++ code | 125M |
| VulBERTa-CNN | Encoder-CNN | C/C++ code | 2M |
| Baseline-BiLSTM | BiLSTM | None | 1M |
| Baseline-TextCNN | TextCNN | None | 1M |

CT5: CodeT5, CB: CodeBERT, UC: UnixCoder,
SC2: StarCoder2, CG2.5: CodeGen2.5

Table 9: Training configurations as reported in respective papers

| Configuration | Value |
|---|---|
| Small Model Epochs ($<$7B) | 10 |
| Large Model Epochs (7B) | 4 |
| VulBERTa Pre-training Steps | 500,000 |
| VulBERTa Fine-tuning Epochs | 10 |
| BiLSTM/TextCNN Epochs | 10 |

## F  COMPUTATIONAL AND MEMORY COMPLEXITY ANALYSIS

This section provides a comprehensive analysis of the computational and memory complexity characteristics of our Basilisk Self-Attention mechanism, comparing it systematically against standard self-attention and the original Infini-attention approach (Munkhdalai et al., 2024).

### F.1  STANDARD SELF-ATTENTION PROCESS AND COMPLEXITY

Standard self-attention computes attention scores for all token pairs simultaneously, requiring the following operations:

1. **Query, Key, Value Computation:** $Q = XW_Q$, $K = XW_K$, $V = XW_V$ with complexity $O(3nd^2) = O(nd^2)$

2. **Attention Matrix Computation:** $A = QK^T$ with complexity $O(n^2d)$

3. **Attention Weight Normalization:** softmax$(A)$ with complexity $O(n^2)$

4. **Output Computation:** $O = $ softmax$(A)V$ with complexity $O(n^2d)$

The dominant terms are the attention matrix computation and output computation, yielding overall computational complexity of $O(n^2d + nd^2)$. For typical transformer dimensions where $n > d$ for long sequences, this becomes $O(n^2d)$. Memory complexity is $O(n^2)$ for storing the attention matrix, making this approach computationally prohibitive for long sequences.

### F.2  ORIGINAL INFINI-ATTENTION PROCESS AND COMPLEXITY

The original Infini-attention processes sequences in fixed-size segments while maintaining a compressive memory state:

1. **Segment Processing:** Divide input sequence into segments of size $S$, complexity $O(1)$

2. **Local Attention:** For each segment, compute standard attention with complexity $O(S^2d)$

3. **Memory Retrieval:** Retrieve from compressive memory:

$$A_{\text{mem}} = \frac{(\text{ELU}(Q) + 1)M^T}{(\text{ELU}(Q) + 1)z^T + \epsilon} \tag{18}$$

where $Q$ is $(S \times d)$, $M$ is $(d \times d)$. Matrix operations yield complexity $O(Sd^2)$

4. **Memory Update:** Update compressive state:

$$M \leftarrow M + (\text{ELU}(K)^T + 1)V \tag{19}$$

$$z \leftarrow z + \sum_{i=1}^{S}(\text{ELU}(K_i) + 1) \tag{20}$$

where $K^T$ is $(d \times S)$ and $V$ is $(S \times d)$, yielding $(d \times d)$ update with complexity $O(Sd^2)$

5. **Streaming Output:** Process and discard segment results immediately, complexity $O(1)$

For $n/S$ segments, total computational complexity becomes: $O\left(\frac{n}{S} \times (S^2 d + 2Sd^2)\right) = O(nSd + 2nd^2)$

Memory complexity remains constant at $O(d^2)$ for the compressive memory state.

### F.3 BASILISK SELF-ATTENTION PROCESS AND COMPLEXITY

Our Basilisk Self-Attention modifies the original Infini-attention by accumulating segment outputs for global context integration:

1. **Segment Processing:** Identical segmentation as Infini-attention with complexity $O(1)$
2. **Dual Attention Computation:** For each segment, compute both:
   - Local dot-product attention: $A_{\text{dot}} = \text{softmax}(QK^T/\sqrt{d})V$ with complexity $O(S^2 d)$
   - Memory-based attention using Equation (4) with complexity $O(Sd^2)$
3. **Memory State Update:** Update compressive memory (identical to Infini-attention) with complexity $O(Sd^2)$:

$$M \leftarrow M + (\text{ELU}(K)^T + 1)V \tag{21}$$

$$z \leftarrow z + \sum_{i=1}^{S}(\text{ELU}(K_i) + 1) \tag{22}$$

4. **Output Accumulation:** Store segment outputs in memory with complexity $O(Sd)$ per segment:

$$\text{total\_mem\_outputs} \leftarrow \text{total\_mem\_outputs} + [A_{\text{mem}}] \tag{23}$$
$$\text{total\_attn\_outputs} \leftarrow \text{total\_attn\_outputs} + [A_{\text{dot}}] \tag{24}$$

5. **Global Integration:** After processing all segments, concatenate with complexity $O(nd)$:

$$\text{total\_mem} = \text{concat}(\text{total\_mem\_outputs}) \tag{25}$$
$$\text{total\_attn} = \text{concat}(\text{total\_attn\_outputs}) \tag{26}$$

6. **Adaptive Gating:** Combine outputs using learnable parameter with complexity $O(nd)$:

$$O = \text{sigmoid}(\beta) \odot \text{total\_mem} + (1 - \text{sigmoid}(\beta)) \odot \text{total\_attn} \tag{27}$$

For $n/S$ segments, the total computational complexity becomes: $O\left(\frac{n}{S} \times (S^2 d + 2Sd^2 + Sd)\right) + O(nd) + O(nd) = O(nSd + 2nd^2 + 2nd)$

Memory complexity is $O(nd) + O(d^2)$ due to accumulation of all segment outputs plus the compressive memory state.

### F.4 COMPLEXITY COMPARISON

Table 10 summarizes the complexity characteristics of all three approaches.

**Complexity Analysis:** For standard attention, when processing long sequences where $n \gg d$, the quadratic term $O(n^2 d)$ dominates. For segment-based approaches, the complexity depends on the relationship between segment size $S$, sequence length $n$, and hidden dimension $d$:

Table 10: Detailed Complexity Comparison of Attention Mechanisms

| Mechanism | Computational | Memory | Sequence Limit |
|---|---|---|---|
| Standard Attention | $O(n^2 d + nd^2)$ | $O(n^2)$ | Available memory |
| Infini-attention | $O(nSd + 2nd^2)$ | $O(d^2)$ | Theoretically infinite |
| Basilisk Attention | $O(nSd + 2nd^2 + 2nd)$ | $O(nd) + O(d^2)$ | Available memory |

- When $d^2 \gg Sd$ (i.e., $d \gg S$), both segment-based methods approach $O(nd^2)$ complexity

- With our model parameters ($S = 2048$, $d = 512$), we have $S > d$, so the $nSd$ term remains significant

- For very long sequences where $n$ is much larger than both $S$ and $d$, the $nd^2$ terms eventually dominate

## F.5 MEMORY EFFICIENCY VALIDATION

To validate our theoretical analysis, we conducted empirical memory consumption measurements across different sequence lengths using NVIDIA A100 40GB GPU. Table 11 presents peak memory consumption for our Basilisk Self-Attention compared to standard attention.

Table 11: Empirical Memory Consumption Analysis

| Length Bin | Range | Basilisk Attention | | Standard Attention | |
|---|---|---|---|---|---|
| | | Peak (MB) | Reserved (MB) | Peak (MB) | Reserved (MB) |
| Bin 1 | [0, 16,384] | 1,338 | 1,442 | 32,409 | 39,696 |
| Bin 2 | [16,384, 32,768] | 1,654 | 1,956 | OOM | OOM |
| Bin 3 | [32,768, 65,536] | 2,213 | 2,638 | OOM | OOM |
| Bin 4 | [65,536, 131,072] | 3,322 | 3,848 | OOM | OOM |

OOM: Out of Memory on NVIDIA A100 40GB GPU

The empirical results confirm our theoretical analysis. Basilisk Self-Attention demonstrates near-linear memory scaling, increasing from 1.3GB to 3.3GB across sequence length bins, while standard attention requires 24× more memory for sequences up to 16K tokens and becomes entirely infeasible beyond this threshold.

## F.6 PRACTICAL IMPLICATIONS

The complexity analysis reveals several key insights:

1. **Computational Efficiency:** Our approach maintains identical computational complexity to Infini-attention while enabling more flexible global context modeling compared to the original streaming approach.

2. **Memory Scaling:** The linear memory growth $O(n \times d)$ represents a practical trade-off, allowing processing of extremely long sequences while remaining substantially more memory-efficient than quadratic attention approaches.

3. **Implementation Flexibility:** By accumulating segment outputs, our approach enables various post-processing operations and global context integration strategies that would be difficult to implement in the original streaming framework.

4. **Sequence Length Limitations:** While our approach cannot process theoretically infinite sequences like the original Infini-attention, the practical sequence length limitations are determined by available system memory rather than algorithmic constraints, making it suitable for most real-world applications.

The empirical validation demonstrates that this architectural choice enables processing of sequences that would be computationally infeasible with standard transformer architectures, while maintaining reasonable memory requirements for deployment in practical applications.

## G  ABLATION STUDY: COMBINED DATASET TRAINING

To further evaluate White-Basilisk's performance and investigate the impact of training data composition, we conducted an ablation study using a combined training approach across all datasets. This experiment involved concatenating the training splits from all five datasets (REVEAL, Draper, VulDeepecker, BigVul, and PRIMEVUL) into a single unified training set. During each training epoch, the model was evaluated on the concatenated testing splits from all datasets to monitor for potential overfitting. Final performance metrics were obtained by evaluating the trained model separately on each dataset's designated validation split, ensuring fair comparison with previous results.

### G.1  EXPERIMENTAL SETUP

The model was trained using the same hyperparameters as described in Section 6.2, but with the following data configuration:

- **Training Data:** Combined training splits from all five datasets into a single training set
- **Testing:** Concatenated testing splits from all datasets, used for monitoring training progress
- **Validation:** Individual Validation splits for each dataset, evaluated separately to assess dataset-specific performance

This unified training approach resulted in a significantly larger and more diverse training set, allowing us to investigate how the model performs when exposed to a broader range of vulnerability patterns and coding styles simultaneously.

### G.2  RESULTS AND ANALYSIS

The results of this combined training approach are presented in Table 12.

Table 12: Combined Training Results Across All Datasets

| Dataset | Precision | Recall | F1 | Accuracy |
|---|---|---|---|---|
| REVEAL | 0.416 | 0.551 | 0.470 | 0.888 |
| Draper | 0.568 | 0.532 | 0.549 | 0.948 |
| VulDeepecker | 0.939 | 0.912 | 0.925 | 0.989 |
| BigVul | 0.936 | 0.940 | 0.938 | 0.993 |
| PRIMEVUL | 0.268 | 0.265 | 0.233 | 0.952 |
| Combined Test | 0.643 | 0.585 | 0.613 | 0.956 |

The model maintains strong performance across most datasets, with particularly robust results on VulDeepecker (F1: 0.925) and BigVul (F1: 0.938), suggesting effective transfer learning across different vulnerability detection tasks. Performance varies significantly across datasets, from an F1 score of 0.938 on BigVul to 0.233 on PRIMEVUL, indicating that some vulnerability patterns may be more challenging to learn in a combined setting. The model maintains high accuracy across all datasets (0.888-0.993), demonstrating robust overall classification performance even with the increased complexity of the combined training task. The model generally maintains a good balance between precision and recall, with some datasets showing nearly identical values (e.g., BigVul: 0.936/0.940), suggesting stable learning of vulnerability patterns.

### G.3  MODEL ROBUSTNESS ANALYSIS

A particularly noteworthy aspect of these results is White-Basilisk's ability to maintain stable performance across multiple diverse datasets without experiencing catastrophic forgetting or overfitting. This is especially significant given the model's relatively compact size of 200M parameters. Several factors contribute to this robustness. The model shows signs of positive transfer learning, where knowledge gained from one dataset appears to benefit the detection of vulnerabilities in others. This is evidenced by the maintenance of high accuracy scores across all datasets despite their varying characteristics. Additionally, the model maintains performance across datasets of different sizes and complexity levels, from the smaller REVEAL dataset to the larger PRIMEVUL dataset. This stability suggests that the model's learning capacity is well-matched to the task complexity.

### G.4 COMPARISON WITH INDIVIDUAL TRAINING

When compared to the individual training results presented in Section 5, the combined training approach shows some interesting trade-offs. For some datasets (VulDeepecker, BigVul), the performance remains close to individual training results, suggesting that the model can effectively learn and maintain dataset-specific patterns even in a combined setting. Performance on more challenging datasets like PRIMEVUL shows some degradation, indicating that the increased diversity of the training data may make it harder for the model to capture some of the more nuanced vulnerability patterns specific to certain datasets. The overall combined test metrics (F1: 0.613, Accuracy: 0.956) demonstrate that White-Basilisk can effectively learn from multiple datasets simultaneously while maintaining reasonable performance across all of them.

This ability to maintain stable performance across diverse datasets without overfitting or experiencing catastrophic forgetting is particularly notable for a model of this size. It suggests that White-Basilisk's architecture strikes an effective balance between model capacity and efficiency, enabling robust multi-task learning without requiring the massive parameter counts typically associated with such capabilities. This finding has important implications for the development of efficient, multi-purpose vulnerability detection systems that can be deployed in resource-constrained environments while maintaining high performance across a range of vulnerability types.

## H ABLATION STUDY: ATTENTION MECHANISMS AND LONG-RANGE VULNERABILITY DETECTION

To thoroughly evaluate White-Basilisk's performance and validate our architectural choices, we conducted a comprehensive ablation study focusing on two key aspects: (1) the effectiveness of our linear-complexity Infini-attention mechanism compared to standard self-attention, and (2) the model's performance across varying sequence lengths. This analysis is particularly important given the prevalence of vulnerabilities that span multiple functions or files, requiring models to maintain effectiveness over long code sequences.

### H.1 EXPERIMENTAL SETUP

**Model**: We used the White-Basilisk checkpoint that was trained on the combined datasets from G. We categorized sequences into 6 length bins for analysis.

### H.2 PERFORMANCE ANALYSIS ACROSS SEQUENCE LENGTHS

Table 13 presents a comprehensive analysis of performance across different sequence lengths and datasets, organized into six bins based on token length ranges.

Several significant findings emerge from this comprehensive analysis that provide insights into the effectiveness of our novel Basilisk self-attention mechanism across varying sequence lengths. The results reveal a compelling pattern where performance often improves with increasing sequence length, contrary to the typical expectation of degradation in long-sequence processing tasks.

The counterintuitive improvement in performance with longer sequences demonstrates that our attention mechanism effectively leverages increased contextual information. Short sequences in Bin 1, while computationally efficient to process, may lack sufficient context for the model to make optimal predictions. This is particularly evident in PRIMEVUL, where F1 scores improve from 0.111 in Bin 1 to 0.411 in Bin 5. The lower F1 score in shorter ranges likely reflects the fundamental challenge that limited context provides insufficient information for definitive classification, leading to more conservative predictions.

Most significantly, the substantial performance improvements occur precisely at the 2048-token threshold where our Basilisk self-attention mechanism becomes active. This pattern is exemplified in BigVul, where the model maintains exceptional performance through longer bins. Similarly, VulDeePecker shows remarkable recovery at Bin 5 (F1=0.857) after declining in middle ranges. This correlation between the activation threshold of our novel attention mechanism and improved performance provides empirical validation of our architectural design, demonstrating that the segmented

Table 13: Performance Analysis Across Sequence Length Bins

| Dataset | Bin | Length Range | Performance Metrics | | | | Class Distribution | | | Avg Length |
|---|---|---|---|---|---|---|---|---|---|---|
| | | | F1 | Precision | Recall | Accuracy | Total | Vuln | Non-Vuln | |
| BigVul | Bin 1 | [0, 512] | 0.921 | 0.933 | 0.909 | 0.993 | 15,879 | 692 | 15,187 | 154 |
| | Bin 2 | [512, 1024] | 0.983 | 0.975 | 0.990 | 0.996 | 1,829 | 200 | 1,629 | 709 |
| | Bin 3 | [1024, 2048] | 0.996 | 1.000 | 0.991 | 0.999 | 801 | 116 | 685 | 1,401 |
| | Bin 4 | [2048, 4096] | 0.957 | 0.930 | 0.985 | 0.976 | 252 | 67 | 185 | 2,816 |
| | Bin 5 | [4096, 16384] | 0.939 | 0.886 | 1.000 | 0.957 | 92 | 31 | 61 | 6,938 |
| | Bin 6 | [16384, 131072] | 0.857 | 0.750 | 1.000 | 0.909 | 11 | 3 | 8 | 30,581 |
| PRIMEVUL | Bin 1 | [0, 512] | 0.111 | 0.105 | 0.117 | 0.975 | 19,700 | 265 | 19,435 | 178 |
| | Bin 2 | [512, 1024] | 0.192 | 0.160 | 0.239 | 0.911 | 3,225 | 142 | 3,083 | 720 |
| | Bin 3 | [1024, 2048] | 0.357 | 0.324 | 0.399 | 0.869 | 1,569 | 143 | 1,426 | 1,416 |
| | Bin 4 | [2048, 4096] | 0.320 | 0.284 | 0.367 | 0.785 | 650 | 90 | 560 | 2,763 |
| | Bin 5 | [4096, 16384] | 0.411 | 0.341 | 0.518 | 0.689 | 267 | 56 | 211 | 7,336 |
| | Bin 6 | [16384, 131072] | 0.250 | 0.200 | 0.333 | 0.684 | 19 | 3 | 16 | 24,020 |
| VulDeePecker | Bin 1 | [0, 512] | 0.955 | 0.961 | 0.948 | 0.994 | 14,093 | 865 | 13,228 | 149 |
| | Bin 2 | [512, 1024] | 0.645 | 0.567 | 0.747 | 0.950 | 1,295 | 79 | 1,216 | 705 |
| | Bin 3 | [1024, 2048] | 0.387 | 0.545 | 0.300 | 0.956 | 427 | 20 | 407 | 1,393 |
| | Bin 4 | [2048, 4096] | 0.167 | 0.200 | 0.143 | 0.932 | 147 | 7 | 140 | 2,734 |
| | Bin 5 | [4096, 16384] | 0.857 | 0.750 | 1.000 | 0.979 | 48 | 3 | 45 | 6,040 |
| | Bin 6 | [16384, 131072] | 0.000 | 0.000 | 0.000 | 1.000 | 5 | 0 | 5 | 45,345 |
| Draper | Bin 1 | [0, 512] | 0.555 | 0.602 | 0.515 | 0.957 | 103,478 | 5,421 | 98,057 | 218 |
| | Bin 2 | [512, 1024] | 0.591 | 0.618 | 0.566 | 0.908 | 21,046 | 2,478 | 18,568 | 701 |
| | Bin 3 | [1024, 2048] | 0.601 | 0.649 | 0.560 | 0.911 | 2,913 | 350 | 2,563 | 1,215 |
| | Bin 4 | [2048, 4096] | 0.750 | 1.000 | 0.600 | 0.947 | 38 | 5 | 33 | 2,398 |
| | Bin 5 | [4096, 16384] | 0.000 | 0.000 | 0.000 | 1.000 | 1 | 0 | 1 | 4,738 |

attention approach successfully captures long-range dependencies that are crucial for complex classification tasks.

The segmented attention approach proves particularly effective for sequences exceeding 2048 tokens, where traditional attention mechanisms typically struggle with computational complexity and memory constraints. The consistent high performance across longer bins demonstrates that our mechanism successfully maintains both efficiency and effectiveness as sequence length increases. This capability is particularly valuable for applications requiring analysis of extended sequential data, whether in natural language processing, code analysis, or other domains involving long-context dependencies.

However, it is important to acknowledge the limitation imposed by the natural distribution of sequence lengths in real-world datasets. The number of samples in the highest token length bins is considerably limited—with only 11 samples in BigVul Bin 6, 19 in PRIMEVUL Bin 6, and 5 in VulDeePecker Bin 6. This sample scarcity constrains the statistical power of our validation for ultra-long sequences and limits our ability to draw definitive conclusions about model performance at the extreme end of the length spectrum. While the observed results are encouraging and demonstrate the mechanism's potential, more extensive evaluation on larger collections of long-sequence samples would strengthen the validation of our architectural capabilities for processing sequences exceeding 16K tokens.

### H.2.1 Full-File PRIMEVUL Performance Analysis

This experiment analyzed complete C/C++ source files from PRIMEVUL rather than isolated vulnerable functions. We cross-matched vulnerable samples from the validation set with their corresponding complete source files and performed classification on entire files up to 524K tokens. Note that PRIMEVUL only provides complete C/C++ files for vulnerable samples, not for clean samples, which explains the distribution patterns observed.

Table 14: Full-File PRIMEVUL Performance Across Sequence Length Bins

| Bin | Length Range | Samples | F1 | Precision | Recall | Accuracy | Vuln% | Avg Length |
|---|---|---|---|---|---|---|---|---|
| Bin 1 | [0, 16384] | 23,713 | 0.244 | 0.177 | 0.390 | 0.962 | 1.6 | 524 |
| Bin 2 | [16384, 32768] | 109 | 0.492 | 0.889 | 0.340 | 0.394 | 86.2 | 23,738 |
| Bin 3 | [32768, 65536] | 75 | 0.520 | 1.000 | 0.351 | 0.360 | 98.7 | 45,178 |
| Bin 4 | [65536, 131072] | 38 | 0.417 | 1.000 | 0.263 | 0.263 | 100.0 | 86,464 |
| Bin 5 | [131072, 262144] | 11 | 0.167 | 1.000 | 0.091 | 0.091 | 100.0 | 158,538 |
| Bin 6 | [262144, 524288] | 2 | 0.667 | 1.000 | 0.500 | 0.500 | 100.0 | 288,283 |

The data distribution reflects PRIMEVUL's structure: Bin 1 contains both vulnerable and clean samples (1.6

F1 scores peak in Bins 2-3 (0.492-0.520) where sufficient context combines with adequate sample sizes. Perfect precision (1.000) emerges for sequences exceeding 32K tokens, but recall deteriorates with length, indicating conservative predictions on longer sequences. The small sample sizes in longer bins (11 samples in Bin 5, 2 in Bin 6) limit statistical confidence for ultra-long sequences, though maintained precision demonstrates the model's capability to process extended context without prediction quality degradation.

# I  ABLATION STUDY: CWE-SPECIFIC PERFORMANCE ANALYSIS

To provide deeper insights into White-Basilisk's vulnerability detection capabilities, we conducted a comprehensive analysis of its performance across different Common Weakness Enumeration (CWE) categories. This analysis focuses on the BigVul, Vuldeepecker and Draper dataset, which provide detailed CWE-level metrics, allowing us to understand the model's strengths and limitations across various vulnerability types.

## I.1  EXPERIMENTAL SETUP

- **Model**: We used the White-Basilisk checkpoint that was trained on the combined datasets from G.

- **Evaluation Splits**: Validation split of each dataset

### I.1.1  DATASET-SPECIFIC PERFORMANCE PATTERNS

**Draper Dataset Performance**    In the Draper dataset (Table 16), we observe a consistent pattern of high precision (1.000) across all CWE categories, but with varying recall rates. CWE-119 (Buffer Overflow) achieves the highest recall (0.597) and F1 score (0.748). CWE-120 (Buffer Copy without Checking Size) shows similar performance (recall=0.592, F1=0.744). CWE-469 and CWE-476 demonstrate progressively lower recall (0.563 and 0.522 respectively). This pattern suggests that while the model is highly precise in its predictions, it exhibits some conservatism in vulnerability detection, particularly for less frequent vulnerability types.

**VulDeePecker Dataset Analysis**    The VulDeePecker results (Table 17) show more balanced precision-recall characteristics. CWE-119 demonstrates near-perfect balance (precision=0.939, recall=0.940). CWE-399 (Resource Management Errors) shows lower but consistent performance (precision=0.776, recall=0.785). The balanced metrics suggest more robust learning of these vulnerability patterns, possibly due to better representation in the training data.

**BigVul Dataset Insights**    The BigVul dataset (Table 18) provides the most comprehensive view of White-Basilisk's capabilities across 50+ CWE categories. Several significant patterns emerge.

Regarding perfect detection cases, 22 CWE categories achieve perfect scores (F1=1.000). These include critical vulnerabilities such as CWE-787 (Out-of-bounds Write) and CWE-310 (Cryptographic Issues); access control issues including CWE-732 (Permission Assignment) and CWE-284 (Access Control); and various severity levels ranging from CWE-59 (Link Following) to CWE-617 (Reachable Assertion). It is notable that perfect detection spans both frequent (over 200 samples) and rare (under 50 samples) categories.

For high-volume vulnerability performance, CWE-119 (2,746 samples) shows excellent performance (F1=0.969), CWE-264 (1,240 samples) demonstrates strong results (F1=0.925), and CWE-20 (1,977 samples) exhibits robust detection (F1=0.933).

Performance degradation patterns are observed in certain vulnerability types. Resource-related vulnerabilities show more variable performance, with CWE-404 (Resource Shutdown) having the lowest F1 score (0.571) and CWE-772 (Missing Release) demonstrating lower precision (0.750). Format-string vulnerabilities (CWE-134) show precision-recall imbalance (0.500/1.000).

The impact of sample size reveals that large sample categories (over 1000 samples) show consistently strong but not perfect performance. Medium-sized categories (100-1000 samples) demonstrate more variable results. Small categories (under 100 samples) often show perfect or near-perfect scores, suggesting potential overfitting.

### I.1.2 CROSS-DATASET PERFORMANCE ANALYSIS

The model's behavior across datasets reveals important patterns. Regarding CWE-119 consistency, as the only vulnerability type present across all three datasets, it shows interesting variation: BigVul with F1=0.969 (balanced precision-recall), VulDeePecker with F1=0.940 (balanced precision-recall), and Draper with F1=0.748 (high precision, lower recall). This variation suggests dataset-specific characteristics affect detection performance. For scale effects, larger datasets (BigVul) generally show more balanced precision-recall trade-offs compared to smaller datasets.

### I.1.3 IMPLICATIONS AND INSIGHTS

These results yield several important insights for vulnerability detection. Regarding architecture effectiveness, White-Basilisk's strong performance across numerous CWE categories validates its hybrid architecture design for vulnerability detection.

Detection patterns indicate that memory-related vulnerabilities consistently show strong detection rates, resource management vulnerabilities present more challenges, and access control vulnerabilities demonstrate surprisingly robust detection.

Practical implications of these findings include that high precision across most categories suggests low false positive rates, variable recall in some categories indicates potential for missed vulnerabilities, and perfect detection in rare categories warrants further investigation for potential overfitting.

| CWE | Description | BigVul | | Draper | | VulDeePecker | |
|---|---|---|---|---|---|---|---|
| | | Samples | F1 | Samples | F1 | Samples | F1 |
| CWE-119 | Buffer Overflow: Classic buffer overflow vulnerabilities | 2,746 | 0.969 | 2,419 | 0.748 | 10,419 | 0.940 |
| CWE-399 | Resource Management Errors: Failures in managing system resources | 1,435 | 0.923 | - | - | 5,596 | 0.780 |
| CWE-20 | Input Validation: Improper input validation | 1,977 | 0.933 | - | - | - | - |
| CWE-264 | Access Control: Permissions, privileges, and access controls | 1,240 | 0.925 | - | - | - | - |
| CWE-120 | Buffer Copy: Buffer copy without checking size of input | - | - | 4,750 | 0.744 | - | - |
| CWE-476 | NULL Pointer Dereference | 501 | 0.971 | 1,208 | 0.686 | - | - |
| CWE-416 | Use After Free: Using memory after it has been freed | 963 | 0.958 | - | - | - | - |
| CWE-200 | Information Exposure: Exposure of sensitive information | 883 | 0.944 | - | - | - | - |

Table 15: Comparison of Most Frequent CWEs Across Datasets

| CWE | Precision | Recall | F1 | Accuracy | Total | Pos. Ratio | Neg. Ratio |
|---|---|---|---|---|---|---|---|
| CWE-119 | 1.000 | 0.597 | 0.748 | 0.597 | 2,419 | 1.000 | 0.000 |
| CWE-120 | 1.000 | 0.592 | 0.744 | 0.592 | 4,750 | 1.000 | 0.000 |
| CWE-469 | 1.000 | 0.563 | 0.721 | 0.563 | 252 | 1.000 | 0.000 |
| CWE-476 | 1.000 | 0.522 | 0.686 | 0.522 | 1,208 | 1.000 | 0.000 |
| CWE-other | 1.000 | 0.472 | 0.642 | 0.472 | 3,579 | 1.000 | 0.000 |
| Overall | 0.609 | 0.532 | 0.568 | 0.948 | 127,476 | 0.065 | 0.935 |

Table 16: Draper Dataset Metrics for All CWEs

| CWE | Precision | Recall | F1 | Accuracy | Total | Pos. Ratio | Neg. Ratio |
|---|---|---|---|---|---|---|---|
| CWE-119 | 0.939 | 0.940 | 0.940 | 0.991 | 10,419 | 0.077 | 0.923 |
| CWE-399 | 0.776 | 0.785 | 0.780 | 0.986 | 5,596 | 0.031 | 0.969 |
| Overall | 0.910 | 0.913 | 0.911 | 0.989 | 16,015 | 0.061 | 0.939 |

Table 17: VulDeePecker Dataset Metrics for All CWEs

| CWE | Prec. | Rec. | F1 | Acc. | Total | Pos. % | Neg. % |
|---|---|---|---|---|---|---|---|
| CWE-119 | 0.978 | 0.960 | 0.969 | 0.995 | 2746 | 8.27 | 91.73 |
| CWE-20 | 0.905 | 0.963 | 0.933 | 0.992 | 1977 | 5.51 | 94.49 |
| CWE-399 | 0.957 | 0.892 | 0.923 | 0.992 | 1435 | 5.16 | 94.84 |
| CWE-264 | 0.925 | 0.925 | 0.925 | 0.994 | 1240 | 4.27 | 95.73 |
| CWE-416 | 0.944 | 0.971 | 0.958 | 0.997 | 963 | 3.63 | 96.37 |
| CWE-200 | 0.913 | 0.977 | 0.944 | 0.994 | 883 | 4.87 | 95.13 |
| CWE-125 | 0.984 | 0.984 | 0.984 | 0.997 | 794 | 7.68 | 92.32 |
| CWE-189 | 0.921 | 0.946 | 0.933 | 0.993 | 695 | 5.32 | 94.68 |
| CWE-362 | 0.939 | 0.861 | 0.899 | 0.988 | 592 | 6.08 | 93.92 |
| CWE-476 | 0.944 | 1.000 | 0.971 | 0.998 | 501 | 3.39 | 96.61 |

Table 18: Top 10 CWEs for BigVul by Sample Size

## J    LARGE LANGUAGE MODEL USAGE

In accordance with ICLR 2026 guidelines, we disclose the use of Large Language Models (LLMs) in the preparation of this manuscript. LLMs were employed exclusively as text editing and rewriting tools to improve the clarity, grammar, and overall presentation of the written content.

Specifically, LLMs were used for rewriting sentences and paragraphs for improved clarity and readability, grammar correction and stylistic improvements, and reformulating existing text to enhance flow and coherence.

We explicitly state that LLMs were **not** used for research ideation or conceptualization, brainstorming or generating research ideas, designing or conducting experiments, data analysis or interpretation, creating novel content, arguments, or scientific contributions, code implementation or algorithm development, or literature review and citation generation.

All research ideas, experimental design, data analysis, results interpretation, and scientific contributions presented in this work are entirely the intellectual product of the authors. The LLMs served purely as editing companions to refine the presentation of our original work. We take full responsibility for all content in this manuscript, including any text that was refined using LLM assistance.

