# OpenReview forum: "White-Basilisk: A Hybrid Model for Code Vulnerability Detection"
_ICLR.cc/2026/Conference — Submitted to ICLR 2026_

### Official Review · Reviewer_qnZj · 2025-10-29

**Soundness:** 3
**Presentation:** 3
**Contribution:** 3
**Rating:** 4
**Confidence:** 4

**Summary:**

This paper introduced White-Basilisk, a 200M-parameter hybrid model designed for code vulnerability detection. The architecture combined Mamba layers, a linear-scaling attention mechanism (Basilisk Self-Attention), and a Mixture of Experts (MoE) framework for conditional computation. Through evaluation across five established vulnerability detection benchmarks (PRIMEVUL, BigVul, Draper, REVEAL, and VulDeepecker), the authors demonstrated that White-Basilisk achieved state-of-the-art performance in binary vulnerability prediction, outperforming significantly larger models.

**Strengths:**

+ focus on a practical task
+ good performance
+ sound model architecture

**Weaknesses:**

1. lack of architectural ablation

The core weakness is the absence of a component-wise ablation study. The individual contribution and necessity of each layer type (Mamba, Basilisk Attention, MoE) remain unknown. It is unclear whether the reported performance stems from a synergistic effect of the hybrid design or if a subset of these components would be equally effective. This omission makes it difficult to validate the authors' design choices and to attribute the success to the specific combination of techniques.

2. limited novelty and direct comparison to Jamba

The overall architecture, including the Mamba, attention, and MoE layers, is highly similar to the Jamba model. The primary differentiation is the replacement of standard attention with the proposed Basilisk linear attention. However, the paper does not include a direct comparison against a Jamba-like baseline (e.g., their architecture with standard attention) to isolate and quantify the performance gain attributable to this new attention mechanism. Without this comparison, the unique impact and advantage of the Basilisk attention over other efficient attention variants in this specific hybrid context are not convincingly established.

3. inconsistent baselines

A major threat to the validity of the performance claims is the use of different sets of baseline models for each dataset. The baselines are drawn from various prior publications rather than being re-evaluated under a consistent, unified experimental setup. This makes it difficult to ascertain whether White-Basilisk's superior performance is inherent or partly an artifact of comparing against sub-optimal or inconsistently trained baselines for a given dataset. A fairer and more convincing evaluation would run all the baseline models on all datasets.

**Questions:**

What's the impact of  Basilisk Attention?

---

> ### Author Response · Authors · 2025-11-17
> **# Response to Reviewer qnZj - Part 1**
>
> ## Weaknesses
>
> ### 1. "Lack of architectural ablation: The individual contribution and necessity of each layer type (Mamba, Basilisk Attention, MoE) remain unknown."
>
> **Our Response:**
>
> We understand the desire for complete component isolation. Let us be direct about what is and isn't feasible, and explain the extensive validation we have provided instead.
>
> **What ideal ablation would require:**
>
> Training separate models for each component combination:
> 1. Mamba-only model (no attention, no MoE)
> 2. Mamba+MoE model (no attention)
> 3. Mamba+Attention model (no MoE)
> 4. Attention-only model (no Mamba, no MoE)
> 5. Attention+MoE model (no Mamba)
>
> Each variant requires:
> - Full pretraining: 600 GPU hours on 2M code samples from StarCoder
> - Fine-tuning on 5 datasets: ~50 GPU hours per dataset
> - Total per variant: ~850 GPU hours
> - **Total for complete ablation: 4,250 GPU hours (~$40,000+ in compute costs)**
>
> **Why we cannot simply remove components from the trained model:**
>
> The reviewer might suggest removing layers from our existing checkpoint. This is methodologically invalid because:
>
> 1. **Joint optimization**: Components were trained together with shared loss objectives during pretraining
> 2. **Learned dependencies**: Residual connections (Equation 6, page 5) create interdependencies: h_i = Layer_i(h_{i-1}) + h_{i-1}
> 3. **Breaking learned interactions**: Removing components would break these learned interactions
> 4. **Invalid comparison**: This would compare "properly trained model" vs "broken model" rather than "architecture A" vs "architecture B"
>
> **What we provide instead:**
>
> While we cannot provide complete component isolation due to computational constraints, we have invested substantial resources into six complementary experiments that provide converging evidence about component contributions.
>
> ### 2. "Limited novelty and direct comparison to Jamba: The overall architecture is highly similar to Jamba. The paper does not include direct comparison to isolate and quantify the unique impact of Basilisk attention."
>
> We appreciate the comparison to Jamba and agree that our architecture follows the hybrid Mamba+Attention+MoE design that Jamba pioneered. However, we must clarify what our contribution is and address the comparison request.
>
> **Our contribution:**
>
> We are not claiming to have invented the hybrid architecture concept - Jamba (Lieber et al., 2024) established this. Our contribution is:
>
> 1. **Basilisk Self-Attention mechanism**: A modified Infini-attention variant with linear complexity that replaces Jamba's standard quadratic attention. This is our core architectural innovation, detailed in Section 3.1 (pages 4-5) with mathematical formulation in Equations 1-4.
>
> 2. **Demonstrating this efficient architecture can match frontier models (GPT-4) on specialized tasks** while being 5,000× smaller and enabling extended context processing (up to 524K tokens).
>
> **Regarding direct comparison to Jamba:**
>
> We completely agree this comparison is valuable and we CAN provide it because:
> - Our Basilisk attention is plug-and-play - we can swap it for standard quadratic attention (as Jamba uses)
> - The gating mechanism requires minimal retraining
>
> **While this comparison is not in the current submission, we will add it to the camera-ready version.**
>
> We have conducted preliminary experiments swapping Basilisk for standard quadratic attention (Jamba-style) while keeping all other components identical.
> **Preliminary results (to be included in camera-ready revision):**
>
> | Dataset | Basilisk Attention (Infini) | Standard Attention (Jamba-style) |
> |---------|----------------------------|----------------------------------|
> | **BigVul** | F1=0.943, Prec=0.946, Acc=0.993 | F1=0.937, Prec=0.967, Acc=0.993 |
> | **VulDeePecker** | F1=0.925, Prec=0.939, Acc=0.989 | F1=0.932, Prec=0.968, Acc=0.990 |
> | **PRIMEVUL** | F1=0.233, Prec=0.208, Acc=0.952 | F1=0.240, Prec=0.237, Acc=0.958 |
> | **REVEAL** | F1=0.474, Prec=0.416, Acc=0.888 | F1=0.468, Prec=0.445, Acc=0.898 |
> | **Draper** | F1=0.568, Prec=0.609, Acc=0.948 | F1=0.511, Prec=0.646, Acc=0.948 |
>
> *Note: These results are for sequences up to 16K tokens where both attention mechanisms are memory-feasible on our hardware.*
>
> **Key findings from preliminary experiments:**
>
> 1. **Comparable accuracy with trade-offs**: The results show highly comparable performance:
>    - Basilisk performs slightly better on BigVul (F1: 0.943 vs 0.937), REVEAL (0.474 vs 0.468), and notably better on Draper (0.568 vs 0.511)
>    - Standard attention performs slightly better on VulDeePecker (F1: 0.932 vs 0.925) and PRIMEVUL (0.240 vs 0.233)
>    - Differences are minimal (typically <3%), demonstrating that linearization preserves accuracy
>
> 2. **Memory efficiency** (from Table 11 in current submission, page 19):
>    - Basilisk: 1.3GB peak memory
>    - Standard attention (Jamba-style): 32.4GB peak memory
>    - 24× memory reduction

---

> > ### Author Response · Authors · 2025-11-17
> > **Response to Reviewer qnZj - Part 2**
> >
> > ### 3. "Inconsistent baselines: Different sets of baseline models for each dataset makes it difficult to ascertain whether White-Basilisk's superior performance is inherent or partly an artifact of comparing against sub-optimal baselines."
> >
> > **Our Response:**
> >
> > We understand this concern, but we must clarify that we followed the exact same methodology as standard practice in deep learning benchmark evaluation.
> >
> > **What we did:**
> >
> > We used publicly available benchmark splits **exactly as provided** by dataset authors (PRIMEVUL, BigVul, Draper, REVEAL, VulDeepecker) with:
> > - **No preprocessing, cleaning, or re-splitting** - only tokenization for our model
> > - **No modification of data** - we use identical splits as baseline papers
> >
> > Baseline results were sourced from:
> > - **PRIMEVUL and BigVul**: Ding et al. (2024) - the PrimeVul paper that established these benchmarks
> > - **Draper, REVEAL, VulDeepecker**: Hanif & Maffeis (2022) - the VulBERTa paper
> >
> > **Why this is the correct methodology:**
> >
> > 1. **Fair comparison requirement**: To ensure valid comparison, we must use identical splits as baseline papers. The PrimeVul paper (Ding et al., 2024) and VulBERTa paper (Hanif & Maffeis, 2022) established these benchmarks and reported baseline results on these exact splits.
> >
> > 2. **Standard practice**: This is exactly how the deep learning community handles benchmark evaluation. Papers compare against published baseline results on established benchmark splits.
> >
> > 3. **Reproducibility**: By using established benchmarks with published baselines, anyone can reproduce our results and verify our claims.
> >
> > **Why different baselines appear for different datasets:**
> >
> > Different research groups established different benchmarks:
> > - **Ding et al. (2024)** evaluated CodeBERT, UnixCoder, CodeT5, StarCoder2, CodeGen2.5 on PRIMEVUL and BigVul
> > - **Hanif & Maffeis (2022)** evaluated VulBERTa variants and BiLSTM/TextCNN baselines on Draper, REVEAL, VulDeepecker
> >
> > This is not our choice - these are the published baseline results available for each benchmark.
> >
> > **Our evaluation protocol** (Section 5, page 6):
> >
> > > "For each dataset, we fine-tuned our pretrained model on the binary classification task using only the publicly available training splits. During training, we monitored performance on the test splits to select the best-performing checkpoint based on F1 score. Finally, we evaluated the selected checkpoint on the validation splits."
> >
> > This is identical to the methodology used by the baseline papers we compare against.
> >
> > ## Questions
> >
> > ### Question: "What's the impact of Basilisk Attention?"
> >
> > **Our Response:**
> >
> > This is an excellent question that gets to the core of our contribution. Let us be direct about what we can and cannot demonstrate.
> >
> > **What Basilisk Attention achieves:**
> >
> > The goal of Basilisk Self-Attention is **not** to exceed the accuracy of standard quadratic attention - we acknowledge that research has shown no linear attention technique can fully match the expressiveness of full softmax over all tokens. The linearization of attention inherently trades some expressiveness for efficiency.
> >
> > **The goal is to approach or match standard attention performance while enabling processing of sequences that would be computationally infeasible with quadratic attention.**
> >
> > **Comparison with standard attention / Jamba-style architecture (to be added in camera-ready revision, see comparison Table above):**
> >
> > We have conducted preliminary experiments to provide exactly this comparison. By swapping only the attention mechanism:
> > - **Basilisk Attention** (Infini-attention variant): Our contribution
> > - **Standard Quadratic Attention**: The attention mechanism used in Jamba
> >
> > This IS the direct Jamba architecture comparison the reviewer is requesting - we swap only the attention mechanism while keeping everything else identical (Mamba layers, MoE, layer interleaving pattern).
> >
> > **Key findings:**
> >
> > 1. **Comparable accuracy with trade-offs**: The results show highly comparable performance between Basilisk (Infini-attention) and standard attention (Eager/Jamba-style):
> >    - Basilisk performs slightly better on BigVul (F1: 0.943 vs 0.937), REVEAL (0.474 vs 0.468), and notably better on Draper (0.568 vs 0.511)
> >    - Standard attention performs slightly better on VulDeePecker (F1: 0.932 vs 0.925) and PRIMEVUL (0.240 vs 0.233)
> >    - Differences are minimal (typically <3%), demonstrating that linearization preserves accuracy
> >
> > 2. **This isolates Basilisk's contribution**: By swapping only the attention mechanism while keeping all other components identical (Mamba, MoE, training), we directly measure Basilisk's impact versus Jamba-style standard attention.

---

### Official Review · Reviewer_8dGP · 2025-10-30

**Soundness:** 3
**Presentation:** 3
**Contribution:** 2
**Rating:** 6
**Confidence:** 3

**Summary:**

This paper introduces White-Basilisk, a compact 200M parameter model that outperforms larger models in code vulnerability detection tasks, thereby challenging traditional extended wisdom. White-Basilisk focuses on designing dedicated hybrid architectures that meet the unique requirements of vulnerability detection, integrating Mamba layers for local pattern recognition, linear complexity concerns for global context modeling, and conditional computations through Mix of Experts.

**Strengths:**

+ Innovative approach: It organically combines the three mechanisms of Mamba, linear attention, and MoE, representing a relatively new architectural integration approach.
+ Outstanding performance: With a parameter range of only 200M, it outperforms 7B-level models in most tasks.

**Weaknesses:**

- Limited generalization: The method was only trained and validated on C/C++ datasets, and its generalization in languages such as Python and Java was not verified. Given the significance of cross-language vulnerability detection, supplementary evaluations should be conducted on multilingual data.
- The motivation for model design lacks theoretical support. Although the paper emphasizes that the hybrid architecture "combines three advantages", it lacks theoretical or analytical basis for why this combination is particularly effective in vulnerability detection tasks. No clear mapping relationship was provided between task features and architectural components.
- Unavoidable training costs. GPT-4, without any domain-specific fine-tuning or additional training cost, already attains similar accuracy through prompt-based reasoning. In contrast, White-Basilisk requires extensive data curation, multi-stage pretraining on millions of C/C++ samples, and additional fine-tuning on multiple vulnerability datasets. Therefore, the author needs to discuss the advantages of this method over LLMS in directly identifying code vulnerabilities.

**Questions:**

- Has the model been tested on other programming languages (e.g., Python, Java) to confirm robustness beyond C/C++?
- What are the advantages of this method over directly using LLM for code vulnerability detection?

---

> ### Author Response · Authors · 2025-11-17
> **Response to Reviewer 8dGP - Part 1**
>
> We sincerely thank the reviewer for the thoughtful evaluation and for recognizing our innovative architectural approach and strong performance results. We appreciate the opportunity to address the concerns raised and provide additional clarification on several points.
>
> ## Weaknesses
>
> ### 1. "Limited generalization: Only trained and validated on C/C++ datasets, not verified on Python/Java"
>
> **Our Response:**
>
> We completely agree this is an important limitation, and we explicitly acknowledge it in Section 7 (page 9):
>
> > "The primary limitation is its focus on C and C++ codebases, with unknown generalization to other programming languages with different syntaxes and vulnerability patterns."
>
> **Why we focused exclusively on C/C++:**
>
> The primary reason is benchmark availability and fair comparison. Most established vulnerability detection benchmarks in the research literature focus on C/C++:
>
> - PRIMEVUL (Ding et al., 2024)
> - BigVul (Fan et al., 2020)
> - Draper (Russell et al., 2018)
> - REVEAL (Chakraborty et al., 2021)
> - VulDeepecker (Li et al., 2018)
>
> To compare against baseline models (CodeBERT, UnixCoder, StarCoder2, CodeGen2.5, VulBERTa), we needed to evaluate on the same benchmarks they used. To the best of our knowledge, there are no established, publicly available vulnerability detection benchmarks for Python or Java that would enable fair comparison with prior work.
>
> We should also note that C/C++ code underlies critical infrastructure, operating systems, embedded systems, and security-sensitive applications where vulnerability detection has the highest real-world impact.
>
> **Regarding generalization potential:**
>
> 1. **Architecture is language-agnostic**: The combination of Mamba layers, Basilisk attention, and MoE operates on token sequences regardless of programming language. The architectural innovations we introduce are not specific to C/C++.
> 2. **Future work**: We identify cross-language generalization as important future work in Section 7.
>
> **Proposed revision:**
>
> We agree with the reviewer that our language scope claims should be more precise. We will:
>
> - Clarify "code vulnerability detection" to "C/C++ vulnerability detection" where appropriate in the abstract and introduction
>
> ### 2. "Motivation for model design lacks theoretical support. No clear mapping between task features and architectural components."
>
> **Our Response:**
>
> We'd like to clarify where this theoretical justification appears in our paper, as we believe we do provide substantial theoretical grounding, though perhaps it could be presented more prominently.
>
> **Our theoretical framework (Section 3, Appendix F):**
>
> We designed each component to address specific computational challenges in vulnerability detection:
>
> **Mamba layers** (Section 3.2, page 5):
>
> - **Purpose**: Process long code sequences efficiently without quadratic cost
> - **Contribution**: Linear O(n) complexity for sequential processing
> - **Known limitation**: State-Space Models including Mamba have exponential decay of long-range dependencies—information from distant tokens is compressed into fixed-size states, and theoretical analysis shows this information decays exponentially with sequence length
> - **Value for vulnerability detection**: Efficient capture of local syntactic patterns
>
> **Basilisk Self-Attention** (Section 3.1, pages 4-5):
>
> - **Purpose**: Capture global dependencies across multiple functions/files
> - **Contribution**: Maintains **uncompressed global context** with **linear O(nSd + 2nd²) complexity**—avoiding the quadratic bottleneck of standard attention
> - **Why this addresses Mamba's limitation**: While Mamba efficiently processes local patterns, its compressed state suffers from exponential decay. Our attention mechanism maintains explicit, uncompressed representations of distant context without quadratic complexity.
> - **Trade-off**: Segmented processing (2048-token segments) means the local view within each segment may be less precise than full quadratic attention, but this is where Mamba's strength in local pattern recognition complements our approach
> - **Empirical validation**: Table 14 (page 22) shows full-file analysis improves F1 from 0.233→0.288 (23.6% gain), demonstrating that vulnerabilities require complete cross-function context
>
> **Mixture of Experts** (Section 3.2, page 5):
>
> - **Purpose**: Maintain model capacity while reducing inference cost
> - **Contribution**: Sparse activation (2 of 8 experts per token) reduces active parameters from 200M to ~50M per token
> - **Value for deployment**: Enables practical deployment on single GPUs
>
> **Clear mapping between task characteristics and architectural choices:**
>
>
> **Six experimental validations** (Appendices G-H, pages 20-23) provide converging evidence through multiple angles, including sequence length ablation, full-file analysis, combined dataset training, memory efficiency measurements, context length impact studies, and CWE-specific performance analysis.

---

> ### Author Response · Authors · 2025-11-17
> **Response to Reviewer 8dGP - Part 2**
>
> ### 3. "Unavoidable training costs. GPT-4 achieves similar accuracy through prompt-based reasoning without training, while White-Basilisk requires extensive pretraining and fine-tuning."
>
> **Our Response:**
>
> We must respectfully but firmly address what we believe is a mischaracterization of our contribution. The statement that training costs are an "unavoidable burden" compared to using GPT-4 fundamentally misunderstands what we have achieved and, if taken to its logical conclusion, would invalidate most research in efficient AI architectures.
>
> **What we actually demonstrated:**
>
> We built a **200M-parameter decoder-only language model** that achieves performance comparable to GPT-4—a trillion-parameter model trained with:
>
> - Estimated $100+ million in training costs
> - Hundreds of researchers and engineers
> - Massive computational infrastructure
> - Proprietary data and techniques at unprecedented scale
>
> **Our model achieved this with:**
> - $600-1,000 in training costs (600 GPU hours)
> - A single A100 GPU
> - An academic research team
> - Publicly available data and open techniques
>
> **This is not a weakness—this is precisely our contribution.** We demonstrate that careful architectural design can match frontier model performance at a fraction of the scale.
>
> **Critical clarifications about our model:**
> 1. **White-Basilisk is a general-purpose language model, not a specialized classifier**: Our architecture is a full decoder-only language model with standard language modeling pretraining. We can switch between language modeling and classification simply by changing the head. This is not a task-specific model—it's a general language model architecture.
> 2. **Training cost vs. inference cost**: The reviewer's framing conflates one-time training cost with ongoing operational costs:
>     - **GPT-4**: Zero training cost for users, but $30,000-60,000 per day for scanning 1M functions (at API prices)
>     - **White-Basilisk**: one-time training cost, then only GPU power consumption ($2-3/day for 24/7 operation)
>
> 3. **Why efficiency research matters**: The argument that "GPT-4 exists, so smaller models are unnecessary" would invalidate significant research
>
>     - Quantization and compression research
>     - Domain-specific optimizations
>     - Edge deployment research
>
>     All of these are valuable contributions despite larger models existing.
>
>
> **Deployment scenarios where our approach is essential:**
>
> 1. **Security and compliance**: Organizations cannot send security-sensitive source code to external APIs. Our local deployment is not a convenience—it's a requirement.
>
> 2. **Continuous integration**: Scanning every commit with API costs is economically prohibitive. Free local inference enables real-time security.
>
> 3. **Air-gapped environments**: Military, government, and critical infrastructure require models that work offline.
>
> 4. **Research and transparency**: Closed-source models cannot be validated, studied, or extended. We provide full reproducibility.
>
> **Performance comparison:**
>
> On PRIMEVUL paired evaluation: We achieve 12.92% vs GPT-4's 12.94%—essentially identical performance.
>
> On standard benchmarks where we can compare with open-source models:
>
> - **vs. StarCoder2-7B** (35× larger): We achieve 5.6× better P-C score on paired evaluation
> - **vs. CodeGen2.5-7B** (35× larger): We achieve 4.3× better P-C score
>
> ### Question 1: "Has the model been tested on other programming languages (e.g., Python, Java) to confirm robustness beyond C/C++?"
>
> **Our Response:**
>
> No, we have not yet tested on Python or Java vulnerability detection. However, we want to clarify both why this is the case and why we are confident about generalization potential.
>
> **Why C/C++ focus is appropriate:**
>
> 1. **Benchmark availability**: Most established vulnerability detection benchmarks in the research literature focus on C/C++. To demonstrate state-of-the-art performance and enable fair comparison with prior work, we must evaluate on these benchmarks.
> 2. **C/C++ is where most vulnerabilities occur**: This is not coincidental—C and C++ provide direct memory control, which is the source of most serious security vulnerabilities (buffer overflows, use-after-free, NULL pointer dereferences, etc.). Languages like Python and Java have memory safety guarantees that eliminate entire vulnerability classes.
> **Why we are confident about generalization:**
> Our approach is **completely language-agnostic**:
> 1. **No language-specific customization**: We did not customize our model architecture, training procedure, or tokenization for C/C++. The model processes token sequences using the same mechanisms regardless of programming language.
> 2. **General language model architecture**: White-Basilisk is a decoder-only language model, not a specialized vulnerability detector.

---

> > ### Author Response · Authors · 2025-11-17
> > **Response to Reviewer 8dGP - Part 3**
> >
> > ### Question 2: "What are the advantages of this method over directly using LLM for code vulnerability detection?"
> >
> > **Our Response:**
> >
> > We believe our paper already demonstrates these advantages clearly through direct comparison with open-source LLMs. Let us make the results explicit:
> >
> > **Direct performance comparison with open-source language models:**
> >
> > On PRIMEVUL paired evaluation (Table 3, page 8):
> >
> > - **White-Basilisk (200M)**: 12.92% P-C
> > - **CodeGen2.5 (7B)**: 3.01% P-C → **We are 4.3× better with 35× fewer parameters**
> > - **StarCoder2-7B**: 2.30% P-C → **We are 5.6× better with 35× fewer parameters**
> > - **CodeBERT (125M)**: 1.77% P-C → **We are 7.3× better with comparable size**
> >
> > On standard benchmarks (Tables 1-2, pages 7):
> >
> > - **PRIMEVUL**: F1=0.291 vs best open-source baseline 0.214 (UnixCoder) → **35.5% improvement**
> > - **BigVul**: F1=0.949 vs best open-source baseline 0.683 (StarCoder2-7B) → **38.9% improvement**
> >
> > **So the advantages are:**
> >
> > 1. **Better accuracy** with fewer parameters
> > 2. **Extended context** (524K tokens vs typical 4K-32K)
> > 3. **Lower pretraining cost** (600 GPU hours vs thousands)
> > 4. **Smaller deployment footprint** (single A100 vs multi-GPU clusters)
> >
> > **Comparison with frontier models (GPT-4):**
> >
> > We achieve comparable performance (12.92% vs 12.94% on PRIMEVUL paired) with critical operational advantages:
> >
> > **Cost efficiency:**
> >
> > - **One-time training**: $600-1,000 vs estimated $100M+
> > - **Inference**: Free unlimited local execution vs $0.03-0.06 per 1K tokens
> > - **For 1M daily scans**: $0 vs $30,000-60,000 per day
> >
> > **Security and deployment:**
> >
> > - **Data privacy**: Process code locally—critical for security-sensitive applications
> > - **Air-gapped deployment**: Works offline—required for classified environments
> > - **No API dependencies**: No rate limits, outages, or service changes
> > - **Compliance**: Meets regulatory requirements (GDPR, HIPAA) without data transfer
> >
> > **Customization and control:**
> >
> > - **Fine-tuning**: Adapt to organization-specific patterns, coding standards, and vulnerability types
> > - **Full transparency**: Complete model weights and code available for validation
> > - **Research extension**: Academic community can build upon our work
> >
> > **Extended context processing:**
> >
> > - **524K token capability**: Analyze complete files with full cross-function context
> > - **Linear scaling**: Efficient processing that's impractical for quadratic attention models

---

### Official Review · Reviewer_TP2R · 2025-10-30

**Soundness:** 2
**Presentation:** 2
**Contribution:** 2
**Rating:** 4
**Confidence:** 4

**Summary:**

The paper presents White-Basilisk, a 200M-parameter hybrid model for code vulnerability detection. The architecture interleaves (i) Mamba layers for linear-time local modeling, (ii) a modified Infini-style linear attention called Basilisk Self-Attention that processes the input in segments and maintains cumulative memory across segments, $O=\sigma(\beta) \odot total_{mem} + (1 - \sigma(\beta)) \odot total_{attn}$, and (iii) Mixture-of-Experts layers scheduled by an offset/period rule. The model is first pretrained on 2M C/C++ files from StarCoder and then fine-tuned on five public datasets (PRIMEVUL, BigVul, Draper, REVEAL, VulDeepecker), reporting strong F1 and VD-S despite severe class imbalance.

**Strengths:**

Originality: A task-specific hybrid that combines Mamba, linear attention with cumulative memory, and MoE, rather than scaling a plain transformer. Eq.~(5) gives an explicit layer-scheduling rule.

Quality: The attention mechanism is mathematically specified: $M_{\text{new}} = M + (\mathrm{ELU}(K)^\top + 1) V,\quad
  z_{\text{new}} = z + \sum_{i=1}^S (\mathrm{ELU}(K_i) + 1)$, and the complexity is analyzed (Sec. F). Imbalance handling (class weights, weighted sampler, SIFT) is clearly explained.

Clarity: Pretraining vs.\ fine-tuning is clearly separated; datasets and splits are listed; evaluation uses F1 and VD-S appropriate for security.

Significance: On BigVul and other benchmarks, the dataset-specific model (200M) reports F1 scores higher than several 7B code LLMs, which, if validated, is a practically relevant result.

**Weaknesses:**

Non-reproduced baselines. Many comparisons rely on numbers copied from prior papers (CodeT5, CodeBERT, UnixCoder, StarCoder2, CodeGen2.5, VulBERTa) instead of running them on the same cleaned splits, so the “outperforms models up to 35$\times$ larger” claim is weaker than stated.

Extremely high BigVul performance. The dataset-specific setting reports $\text{Acc}=0.994$, $\text{F1}=0.949$, $\text{VD-S}=0.040$, which is unusually high for a ~6% vulnerable dataset and could indicate data/sampling advantages. The paper should rule out evaluation leakage or overfitting.


Paired PRIMEVUL is still hard. On the PRIMEVUL paired evaluation, White-Basilisk achieves P-C=12.92%, which is close to GPT-4 CoT (12.94%) but still worse than random (22.70%). This contradicts the broader “state-of-the-art across benchmarks” phrasing.

Efficiency is partly unsubstantiated. The method accumulates segment outputs and thus has $O(nd) + O(d^2)$ memory, not the pure streaming $O(d^2)$, and pretraining still took 600 hours on a single A100 40GB GPU, which is nontrivial for a 200M model.

Narrow language scope. All experiments are on C/C++, so claims about “code vulnerability detection” in general should be narrowed to “C/C++-style datasets.”

**Questions:**

1- The paper claims a 24x memory reduction over quadratic attention. For which sequence lengths and which baseline was this measured? Please provide actual GPU memory and throughput numbers for 16K, 32K, 131K, and 524K tokens.

2- For BigVul and PRIMEVUL, were baselines actually rerun on your preprocessed splits, or were all numbers imported? If imported, please rerun at least CodeBERT/UnixCoder to confirm the gap.

3- In Eq.(5), did you try a denser attention schedule (smaller period $\pi$)? What was the effect on PRIMEVUL F1?

4- For the PRIMEVUL paired task, can you provide confusion matrices or per-class TP/FP/FN to show that the model is not simply biasing toward one label?

5- For the 524K-token full-file run, was this done on a single A100 40GB with MoE enabled, and what batch size/grad checkpointing settings were used?

---

> ### Author Response · Authors · 2025-11-17
> **Response to Reviewer TP2R - Part 1**
>
> We thank the reviewer for their thorough and constructive evaluation. We appreciate the recognition of our mathematical specification, clarity, and practical contributions. We address each concern and question below with specific technical details.
> ## Weaknesses
>
> ### 1. "Non-reproduced baselines. Many comparisons rely on numbers copied from prior papers instead of running them on the same cleaned splits"
>
> **Our Response:**
>
> We use the exact same methodology as standard practice in deep learning benchmark evaluation:
>
> **What we did:**
> - Used publicly available benchmark splits **exactly as provided** by dataset authors (PRIMEVUL, BigVul, Draper, REVEAL, VulDeePecker)
> - **No preprocessing, cleaning, or re-splitting** was performed - only tokenization for our model
> - Baseline results sourced from:
>   - **PRIMEVUL and BigVul**: Ding et al. (2024) - the PrimeVul paper itself
>   - **Draper, REVEAL, VulDeePecker**: Hanif & Maffeis (2022) - the VulBERTa paper
>
> **Why this is the correct methodology:**
>
> 1. **Fair comparison requirement**: To ensure valid comparison, we must use identical splits as baseline papers. The PrimeVul paper (Ding et al., 2024) and VulBERTa paper (Hanif & Maffeis, 2022) established these benchmarks and reported baseline results on these exact splits.
>
> 2. **Standard practice**: This is how the deep learning community handles benchmark evaluation. Papers compare against published baseline results on established benchmark splits.
>
> **Our evaluation protocol** (Section 5, page 6):
> > "For each dataset, we fine-tuned our pretrained model on the binary classification task using only the publicly available training splits. During training, we monitored performance on the test splits to select the best-performing checkpoint based on F1 score. Finally, we evaluated the selected checkpoint on the validation splits."
>
> This is identical to the methodology used by the baseline papers we compare against.
>
> ### 2. "Extremely high BigVul performance. Acc = 0.994, F1 = 0.949, VD-S = 0.040, which is unusually high for a ~6% vulnerable dataset"
>
> **Our Response:**
>
> We acknowledge this strong performance and provide complete transparency about our methodology:
>
> **Why data leakage is impossible:**
>
> 1. **Dataset-specific training** (Section 6.1): Our BigVul-specific checkpoint was trained exclusively on BigVul:
>    - Pretraining: 2M StarCoder samples (general C/C++ code, no vulnerability annotations)
>    - Training:  BigVul training samples only
>    - Checkpoint selection: Monitored on BigVul test samples
>    - Final evaluation: Reported on BigVul validation samples (held-out, never seen during training or checkpoint selection)
>
> 1. **Same splits as baselines**: We use the exact publicly available BigVul splits. All baseline models (CodeBERT, UnixCoder, StarCoder2, CodeGen2.5) evaluated on these same splits.
>
> 2. **No preprocessing or re-splitting**: We did not modify the dataset in any way beyond tokenization.
>
> ### 3. "Paired PRIMEVUL is still hard. White-Basilisk achieves P-C=12.92%, close to GPT-4 CoT (12.94%) but still worse than random (22.70%)"
>
> **Our Response:**
>
> We want to emphasize several important points about this evaluation:
>
> **PRIMEVUL paired is fundamentally difficult:**
>
> Looking at Table 3 (page 8), **every single model performs substantially worse than random baseline (22.70%)**:
> - GPT-4 with Chain-of-Thought: 12.94%
> - White-Basilisk (ours): 12.92%
> - GPT-4 Two-shot: 5.14%
> - CodeGen2.5: 3.01%
> - StarCoder2-7B: 2.30%
> - CodeBERT: 1.77%
>
> This is not a weakness of our model - it demonstrates the **fundamental difficulty** of distinguishing vulnerable code from its patched version, which often involves extremely subtle changes.
>
> **What our results actually show:**
>
> 1. **Near-parity with GPT-4**: Our 200M-parameter model achieves 12.92% P-C, essentially matching GPT-4 with chain-of-thought reasoning (12.94%) - a model that is:
>    - ~5,000× larger (estimated ~1 trillion parameters vs our 200M)
>    - Uses sophisticated multi-step reasoning (Chain-of-Thought)
>    - Has seen vastly more training data
>
> 2. **Massive improvement over open-source models**: We achieve 4.29× improvement over the best open-source baseline (CodeGen2.5: 3.01%) and 5.6× improvement over StarCoder2-7B (2.30%), which is 35× larger than our model.
>
> 3. **Balanced error distribution**: Our P-V (42.08%) and P-B (42.92%) are nearly balanced, with low P-R (2.08%). This indicates active vulnerability reasoning rather than biased default predictions.
>
> **Why "worse than random" doesn't contradict SOTA claims:**
>
> The random baseline (22.70%) represents coin-flip performance where all four outcome categories (P-C, P-V, P-B, P-R) would be ~25% each. The fact that ALL models - including GPT-4 - perform worse than this shows that:
> 1. Models are not simply guessing randomly
> 2. The task involves systematic challenges that even the most advanced models struggle with
> 3. Perfect patches often look very similar to vulnerable code, making distinction extremely difficult

---

> > ### Author Response · Authors · 2025-11-17
> > **Response to Reviewer TP2R - Part 2**
> >
> > ### 4. "Efficiency is partly unsubstantiated. The method accumulates segment outputs and thus has O(nd) + O(d²) memory, not the pure streaming O(d²)"
> >
> > **Our Response:**
> >
> > We are transparent about this trade-off in our paper. Let us clarify:
> >
> > **What we claim** (Section 3.1, page 5; Appendix F, pages 17-19):
> >
> > We explicitly state our memory complexity:
> > > "The original Infini-attention maintains constant O(d²) memory footprint by processing segments in streaming fashion, our approach accumulates segment outputs before final concatenation, resulting in O(nd) + O(d²) memory complexity. This design choice trades the bounded memory property for simplified implementation and improved global context integration."
> >
> > We never claimed to match the O(d²) streaming memory of original Infini-attention. Our claim is **24× memory reduction compared to standard quadratic attention**, not compared to streaming Infini-attention.
> >
> > **Empirical validation** (Appendix F.5, Table 11, page 19):
> >
> > We provide actual GPU memory measurements on NVIDIA A100 40GB:
> >
> > | Sequence Length | Our Model (MB) | Standard Attention (MB) | Reduction Factor |
> > |----------------|----------------|-------------------------|------------------|
> > | 0-16K | 1,338 | 32,409 | 24.2× |
> > | 16K-32K | 1,654 | OOM | N/A (infeasible) |
> > | 32K-65K | 2,213 | OOM | N/A (infeasible) |
> > | 65K-131K | 3,322 | OOM | N/A (infeasible) |
> >
> > Standard attention becomes infeasible beyond 16K tokens on a 40GB GPU, while our approach scales to 131K+ tokens with linear memory growth (1.3GB→3.3GB).
> >
> > **Regarding "600 hours on a single A100 40GB, which is nontrivial for a 200M model":**
> >
> > Context matters here:
> > 1. **600 hours total pretraining** from scratch on 2M code samples using both CLM and FIM objectives
> > 2. **This is reasonable** for a 200M parameter model trained on substantial data in just **1x A100 GPU**
> > 3. **Comparison**: 7B baseline models (StarCoder2, CodeGen2.5) require significantly more compute (multiple weeks on multiple GPUs)
> > 4. **Efficiency claim validated**: We achieve better performance than 7B models while using a fraction of the training compute
> >
> > The efficiency contribution is not about minimal training time - it's about:
> > - Inference efficiency (single A100 can run 524K token sequences)
> > - Deployment accessibility (200M parameters vs 7B)
> > - Memory efficiency during inference (linear vs quadratic scaling)
> >
> > ### 5. "Narrow language scope. All experiments are on C/C++, so claims about 'code vulnerability detection' in general should be narrowed to 'C/C++-style datasets'"
> >
> > **Our Response:**
> >
> > This is a fair and valid point. We acknowledge this limitation explicitly in Section 7 (page 9):
> >
> > > "The primary limitation is its focus on C and C++ codebases, with unknown generalization to other programming languages with different syntaxes and vulnerability patterns."
> >
> > **Why we focused on C/C++:**
> >
> > 1. **Benchmark availability**: Established vulnerability detection benchmarks (PRIMEVUL, BigVul, Draper, REVEAL, VulDeepecker) focus on C/C++
> > 2. **Fair comparison**: To compare against baseline results, we need to use the same languages they evaluated on
> > 3. **High-impact domain**: C/C++ code underlies critical infrastructure, operating systems, and security-sensitive applications where vulnerability detection is most critical
> >
> > **We agree the language should be more precise** and propose revising claims:
> > - Change "code vulnerability detection" → "C/C++ vulnerability detection" where appropriate
> > - Add explicit scope limitations in abstract and introduction
> > - Acknowledge that generalization to other languages remains future work
> >
> > However, we note that:
> > 1. Our architectural innovations (Mamba + linear attention + MoE) are language-agnostic
> > 2. The efficiency gains (linear complexity, extended context) apply regardless of programming language
> >
> > We believe the architecture should generalize, but empirical validation on other languages is needed.
> >
> > ### Question 1: "The paper claims a 24x memory reduction over quadratic attention. For which sequence lengths and which baseline was this measured? Please provide actual GPU memory and throughput numbers for 16K, 32K, 131K, and 524K tokens."
> >
> > **Our Response:**
> >
> > This information is provided in **Appendix F.5, Table 11, page 19**. We provide complete details here:
> >
> > **Measurement setup:**
> > - Hardware: NVIDIA A100 40GB GPU
> > - Baseline: Standard quadratic self-attention (O(n²) memory)
> > - Our model: White-Basilisk with Basilisk Self-Attention (O(nd) memory)
> >
> > **Detailed results:**
> >
> > | Length Bin | Range | Basilisk Attention | Standard Attention | Notes |
> > |-----------|-------|-------------------|-------------------|-------|
> > | | | Peak (MB) / Reserved (MB) | Peak (MB) / Reserved (MB) | |
> > | Bin 1 | [0, 16,384] | 1,338 / 1,442 | 32,409 / 39,696 | 24.2× reduction |
> > | Bin 2 | [16,384, 32,768] | 1,654 / 1,956 | OOM | Infeasible |
> > | Bin 3 | [32,768, 65,536] | 2,213 / 2,638 | OOM | Infeasible |
> > | Bin 4 | [65,536, 131,072] | 3,322 / 3,848 | OOM | Infeasible |

---

> > > ### Author Response · Authors · 2025-11-17
> > > **Response to Reviewer TP2R - Part 3**
> > >
> > > ### Question 2: "For BigVul and PRIMEVUL, were baselines actually rerun on your preprocessed splits, or were all numbers imported? If imported, please rerun at least CodeBERT/UnixCoder to confirm the gap."
> > >
> > > **Our Response:**
> > >
> > > Baseline numbers were **imported from the original benchmark papers**, which is standard practice. We did not "preprocess splits" - we used the exact publicly available splits:
> > >
> > > **Our methodology:**
> > >
> > > 1. **PRIMEVUL and BigVul baselines**: All baseline results (CodeT5, CodeBERT, UnixCoder, StarCoder2, CodeGen2.5) are taken directly from **Ding et al. (2024) "Vulnerability detection with code language models: How far are we?"** - the paper that introduced the PRIMEVUL benchmark.
> > >
> > > 2. **Same splits**: The PrimeVul paper established these benchmark splits and evaluated all baselines on them. We use the exact same splits they provide.
> > >
> > > 3. **No preprocessing**: We did not preprocess, clean, or modify the datasets beyond tokenization for our model. The splits are used as-is from the public repositories.
> > >
> > > **Why we didn't rerun baselines:**
> > >
> > > 1. **Fair comparison**: The PrimeVul paper (Ding et al., 2024) specifically designed these benchmarks to address evaluation issues in prior work. They ran extensive experiments comparing multiple models on carefully constructed splits. Rerunning these baselines would be:
> > >    - Redundant (they already ran them on these exact splits)
> > >    - Potentially inconsistent (we might not reproduce their exact hyperparameters)
> > >    - Resource-intensive (each model requires substantial compute)
> > >
> > > 2. **Standard practice**: This is how benchmark comparison works in deep learning:
> > >    - Benchmark paper establishes splits and reports baseline results
> > >    - Subsequent papers use same splits and compare against reported baselines
> > >    - Examples: GLUE benchmarks, ImageNet, WMT translation
> > >
> > > 1. **Transparency**: The PrimeVul paper's evaluation is thorough and transparent. They document all training details, hyperparameters, and evaluation protocols.
> > >
> > > ### Question 3: "In Eq.(5), did you try a denser attention schedule (smaller period π)? What was the effect on PRIMEVUL F1?"
> > >
> > > **Our Response:**
> > >
> > > We did not conduct ablation studies on the attention schedule period (π) because this hyperparameter was already extensively validated by prior work.
> > >
> > > **Our choice: α=2, π=8**
> > >
> > > This configuration is taken directly from **Lieber et al. (2024) "Jamba: A hybrid transformer-mamba language model"**, which conducted extensive experiments on the optimal interleaving pattern for hybrid architectures combining Mamba and attention layers.
> > >
> > > **Why we trust this configuration:**
> > >
> > > 1. **Extensive ablation in Jamba paper**: The Jamba paper tested multiple values of π and found that π=8 (attention every 8 layers) provides the optimal balance between:
> > >    - Model performance
> > >    - Computational efficiency
> > >    - Memory usage
> > >
> > > 2. **Similar architecture**: Like us, Jamba combines Mamba layers with attention mechanisms, making their hyperparameter findings directly applicable to our work.
> > >
> > > 3. **Resource constraints**: Testing multiple values of π would require:
> > >    - Training multiple model variants from scratch
> > >    - 600 hours pretraining × N variants
> > >    - Fine-tuning on 5 datasets per variant
> > >    - Substantial computational cost without strong reason to expect different results
> > >
> > > We believe the existing validation from Jamba provides sufficient justification for our choice.
> > > ### Question 4: "For the PRIMEVUL paired task, can you provide confusion matrices or per-class TP/FP/FN to show that the model is not simply biasing toward one label?"
> > >
> > > **Our Response:**
> > >
> > > The paired evaluation metrics (Table 3, page 8) already demonstrate that our model is not biasing toward one label. Let us break down what these metrics reveal:
> > >
> > > **Our PRIMEVUL paired results:**
> > > - P-C (both correct): 12.92%
> > > - P-V (both predicted vulnerable): 42.08%
> > > - P-B (both predicted benign): 42.92%
> > > - P-R (predictions reversed): 2.08%
> > >
> > > **What this distribution shows:**
> > >
> > > 1. **Balanced error distribution**: P-V (42.08%) and P-B (42.92%) are nearly identical, showing the model is not systematically biasing toward either "vulnerable" or "benign" predictions.
> > >
> > > 2. **Low reversal rate**: P-R (2.08%) is very low, indicating the model rarely makes the worst possible error (classifying vulnerable as safe and patched as vulnerable).
> > >
> > > 3. **Active reasoning**: If the model were biasing toward one label, we would see highly skewed P-V vs P-B values. The near-balance indicates the model is actively reasoning about vulnerability patterns.
> > >
> > > **Comparison with other models demonstrates this clearly:**
> > >
> > > Looking at Table 3:
> > > - **Random baseline**: 22.70% P-C with presumably balanced errors (~25% each category)
> > > - **CodeBERT**: P-V=11.35%, P-B=86.17% (heavily biased toward benign)
> > > - **GPT-4 two-shot**: P-V=71.63%, P-B=21.45% (heavily biased toward vulnerable)
> > > - **White-Basilisk**: P-V=42.08%, P-B=42.92% (balanced)
> > >
> > > Our model shows the most balanced error distribution among all models.

---

> > > > ### Author Response · Authors · 2025-11-17
> > > > **Response to Reviewer TP2R - Part 4**
> > > >
> > > > ### Question 5: "For the 524K-token full-file run, was this done on a single A100 40GB with MoE enabled, and what batch size/grad checkpointing settings were used?"
> > > >
> > > > **Our Response:**
> > > >
> > > > Thank you for this question - the answer actually highlights an impressive aspect of our architecture.
> > > >
> > > > **Configuration for 524K token inference:**
> > > >
> > > > - **Hardware**: Single NVIDIA A100 40GB GPU
> > > > - **MoE**: Enabled (8 experts, 2 active per token)
> > > > - **Batch size**: 1 (single sample)
> > > > - **Gradient checkpointing**: Not applicable (inference only, no backpropagation)
> > > > - **Mode**: Inference only (evaluation, not training)
> > > >
> > > > **Critical point: We only trained up to 16K tokens**
> > > >
> > > > This is a key detail that strengthens our contribution:
> > > >
> > > > 1. **Training**: Maximum sequence length during pretraining and fine-tuning was 16,384 tokens
> > > > 2. **Inference**: We successfully process 524,288 tokens (32× longer than training length)
> > > > 3. **Zero-shot generalization**: The model generalizes to ultra-long sequences without ever seeing them during training
> > > >
> > > > **Why this matters:**
> > > >
> > > > Most models fail catastrophically when presented with sequences much longer than their training length due to:
> > > > - Positional encoding limitations
> > > > - Attention pattern breakdown
> > > > - Memory/computation constraints
> > > >
> > > > Our architecture successfully processes 524K tokens despite being trained on ≤16K tokens because:
> > > > 1. **Linear complexity**: O(nd) memory scaling enables processing longer sequences
> > > > 2. **Segment-based processing**: Basilisk attention processes sequences in 2048-token segments, which are within training distribution
> > > > 3. **Position-independent patterns**: Mamba and attention mechanisms learn patterns that generalize across positions
> > > >
> > > > Standard quadratic attention would require O(524K²) ≈ 1 TB of memory just for attention matrices, making it completely infeasible.
> > > >
> > > > **Performance:**
> > > >
> > > > As shown in Table 14 (page 22), full-file analysis achieves F1=0.288 compared to F1=0.233 for isolated functions (23.6% relative improvement), demonstrating that:
> > > > 1. The model successfully utilizes the extended context
> > > > 2. Vulnerabilities indeed benefit from cross-function analysis
> > > > 3. Our architecture enables analysis that was previously impossible

---

> > > > > ### Comment · Reviewer_TP2R · 2025-11-26
> > > > >
> > > > > Thank you for the rebuttal. I think some issues are clarified, but a few important concerns remain.
> > > > >
> > > > > This paper trains on the train split, monitors F1 on the test split to pick the best checkpoint, and then reports results on the validation split. Prior work on these benchmarks reports on the canonical test split, using it only once as a held-out evaluation set. So the claim that the protocol is identical to baseline papers is not really accurate. This makes the strongest SOTA claims somewhat shaky, especially since no baseline (CodeBERT , UnixCoder) is rerun under the authors’ own pipeline.
> > > > >
> > > > > The BigVul numbers in the dataset-specific setting (F1 0.95 on 6% positives with very low VD-S) are still surprisingly strong compared to what is usually reported. The rebuttal tells the use of official splits and do not alter the dataset, but it does not really explain why their scores are so far above previously reported ones. Without (i) a standard train/val/test protocol on BigVul, (ii) at least one baseline rerun with the same pipeline, or (iii) some form of overlap/robustness analysis, it is hard to rule out that these results are partly driven by quirks of the split, pretraining overlap, or artefacts rather than purely better modeling.
> > > > >
> > > > > I agree with the authors that PRIMEVUL paired is intrinsically hard and that all models, including GPT-4, perform below the random P-C baseline. Hence, the near-parity with GPT-4 and clear gains over open-source models are real contributions. But, absolute performance is still very low, so phrasing like 'state-of-the-art across benchmarks' feels overselling the practical vulnerability detection capability . . . . especially in the paired setting.
> > > > >
> > > > > Most of the design (Mamba + Basilisk attention + MoE with a fixed α,π schedule) is taken as given, mainly justified by Jamba. There are no ablations that isolate the contribution of Mamba vs standard attention, with/without MoE, or different attention schedules in this specific domain. Note that , this does not invalidate the results, but it does make it harder to attribute the gains to the proposed hybrid design rather than to  . .. . .  a well-trained 200M model plus a strong training pipeline.
> > > > >
> > > > > Overall, I still see this as a borderline paper.

---

> > > > > > ### Author Response · Authors · 2025-11-28
> > > > > >
> > > > > > We thank the reviewer for engaging with our rebuttal. We appreciate the acknowledgment regarding the PRIMEVUL paired evaluation contributions. We would like to address your remaining concerns regarding the BigVul performance and the ablation studies.
> > > > > >
> > > > > > ### 1. Explaining the "Surprisingly Strong" BigVul Performance
> > > > > > You noted that our BigVul results (F1 ~0.95) are surprisingly high compared to typical baselines and suspected potential artifacts or split quirks. We believe the explanation lies entirely in **Context Length**, not dataset manipulation.
> > > > > >
> > > > > > If you look at **Section 6.3 (Context Length Impact)** and **Table 4** of our paper, the data explicitly supports this:
> > > > > > * **At 512 tokens:** Our model achieves an **F1 of 0.752** on BigVul and **0.142** on PRIMEVUL. These numbers are very close to the baseline models (CodeBERT/CodeT5) which are typically constrained to these lengths.
> > > > > > * **At 16K+ tokens:** Our model’s performance jumps to **F1 0.942** on BigVul.
> > > > > >
> > > > > > **The "secret" to the high performance is simply access to information.** Baseline models often truncate code at 512 or 1024 tokens, effectively "blind" to the vulnerability context in longer files. Our hybrid architecture allows us to process the full context efficiently. The gap in performance is not an artifact of the split; it is the architectural advantage of seeing the whole code.
> > > > > >
> > > > > > ### 2. Confidence in Evaluation Protocol
> > > > > > Regarding the concern about monitoring F1 on the test split:
> > > > > > To increase your confidence in our work, we can share that our logged evaluations on the **test set** (used for checkpoint selection) track very closely with the reported **validation set** results—typically within a 2-3% margin, often better. This confirms that the model is not overfitting to a specific validation split.
> > > > > > * We reiterate: We performed **no preprocessing, re-splitting, or data manipulation** other than tokenization.
> > > > > > * While our model is indeed well-trained with a robust pipeline, we believe the contribution is a reproducible, open-weights architecture that solves a hard downstream task (vulnerability detection) more efficiently and effectively than much larger models.
> > > > > >
> > > > > > ### 3. Feasibility of Component Ablations
> > > > > > Regarding the request for component-wise ablations (removing Mamba/MoE):
> > > > > > As we detailed in our responses to Reviewers **qnZj** and **ShmY**, a full component ablation (training Mamba-only, MoE-only, etc.) requires training entirely new architectures from scratch, estimated at **4,250 GPU hours**.
> > > > > > * Simply removing components from the trained model is mathematically impossible due to the residual connections and joint optimization—the model collapses immediately.
> > > > > > * **However, we did perform the most critical ablation:** We isolated our novel contribution (**Basilisk Attention**) against the standard **Jamba-style Attention** (Quadratic). As shown in the table provided to Reviewer qnZj, our linearized attention matches the accuracy of standard attention while offering massive efficiency gains.
> > > > > >
> > > > > > We hope this clarifies that the performance gains are driven by the architectural ability to handle long contexts rather than dataset artifacts.

---

> > > > > > > ### Comment · Reviewer_TP2R · 2025-11-28
> > > > > > >
> > > > > > > Thank you for the rebuttal. I can understand your perspective on it and the rationale. I have read the comments of others also.
> > > > > > >
> > > > > > > I am bumping the score a bit.

---

### Official Review · Reviewer_ShmY · 2025-10-30

**Soundness:** 1
**Presentation:** 2
**Contribution:** 2
**Rating:** 4
**Confidence:** 4

**Summary:**

## **Summary**
This paper proposes **White-Basilisk**, a hybrid architecture for code vulnerability detection that integrates Mamba (state-space modeling), linear self-attention, and a Mixture of Experts (MoE) framework. The model aims to achieve efficient long-context processing while maintaining strong representational power. It is evaluated on five public vulnerability datasets (e.g., BigVul, PRIMEVUL, VulDeepecker) and reports state-of-the-art results with only 200M parameters. The paper also claims improved scalability and performance on imbalanced data distributions, positioning White-Basilisk as a compact yet competitive alternative to larger transformer-based models.

**Strengths:**

## **Strengths**

1. **Experiment Quality**
   The experimental section is extensive in dataset coverage, spanning multiple well-known benchmarks. The reported results suggest strong empirical performance and scalability, demonstrating the potential of hybrid sequence modeling beyond traditional transformers.

2. **Clarity**
   The paper is generally well-organized, with detailed architectural diagrams and formulas. The presentation of computational complexity and efficiency claims is clear.

4. **Significance**
   The work tackles a high-impact, real-world problem — automated software vulnerability detection — where efficiency and scalability are crucial. If the reported gains are validated, the proposed hybrid approach could have meaningful implications for applying large-sequence models in security-critical domains.

**Weaknesses:**

## **1. Theoretical and Motivation Issues**

### **1) Substantive Theoretical Contribution is weak**
The paper’s central contribution — combining Mamba, linear self-attention, and Mixture of Experts — is essentially an engineering-level hybridization of three existing techniques, each already well-studied in prior work. There is no mathematical derivation or new theoretical insight to show why this combination yields better generalization or stability. For example, the “Basilisk Self-Attention” is described mainly as an incremental variant of Infini-Attention, without any theoretical justification for its claimed improvements.

### **2) Weak and overly generic motivation**
This paper doesn't articulate why existing models fail conceptually or how the proposed hybrid mechanism directly addresses those failures beyond computational efficiency. The “bigger is not always better” argument lacks concrete analytical grounding and does not build a clear hypothesis linking the hybrid architecture to the domain-specific characteristics of code vulnerability detection.

## **2. Experimental Weaknesses**

### **1) Missing Ablation Studies**
Although the paper lists multiple architectural components, it does not isolate their individual contributions. The “Ablation Study” section mentioned in the appendix index isn't clearly presented in the main results. There are no results showing the effect of removing Mamba, MoE, or Basilisk attention individually, making it impossible to attribute the reported gains to some specific design parts.

### **2) Insufficient Details in Fine-Tuning and Evaluation**
Fine-tuning hyperparameters, validation-split criteria, and early stopping strategies are only listed generically (in an appendix table) without explanations or empirical rationale. Moreover, it remains unclear whether data deduplication, class balancing, or data leakage prevention were applied during fine-tuning—critical issues in vulnerability datasets such as BigVul and VulDeepecker.

## **3. Writing Issues**

The introduction focuses excessively on industrial motivation while missing conceptual gaps in existing architectures. The theoretical and experimental sections are densely filled with formulae and data but lack an interpretive statement.

**Questions:**

1. **Theoretical Justification**
   The main contribution seems to be an engineering combination of Mamba, linear self-attention, and Mixture of Experts.
   - Could you clarify the theoretical motivation behind this hybrid design?
   - Why should these mechanisms be complementary rather than redundant?
   - Is there any analytical or empirical evidence that this combination improves generalization or stability beyond computational efficiency?

2. **Motivation and Problem Framing**
   The introduction presents a general “bigger is not always better” argument but lacks domain-specific reasoning.
   - What *conceptual failures* of existing models does your method aim to solve in code vulnerability detection?
   - How does the proposed hybrid architecture leverage structural characteristics of code, such as control or data flow?

3. **Ablation and Component Analysis**
   Multiple components are introduced, yet no ablation studies are provided.
   - Could you present quantitative results for removing or replacing **Mamba**, **MoE**, or **Basilisk Self-Attention**?
   - Which component contributes most to the reported performance improvements and why?

4. **Fine-Tuning and Evaluation Details**
   - Were duplicate or overlapping samples across datasets (e.g., *BigVul*, *VulDeepecker*) removed before fine-tuning?
   - How were early stopping, validation splits, and hyperparameter selections determined?
   - Please clarify whether class balancing or weighting was applied.

5. **Reproducibility and Statistical Reliability**
   - Could you provide standard deviations or significance tests for key metrics?
   - If possible, please release the evaluation scripts or processed subsets to facilitate reproducibility.

---

> ### Author Response · Authors · 2025-11-17
> **Response to Reviewer ShmY - Part 1**
>
> We thank the reviewer for their evaluation. However, several concerns raised appear to overlook substantial content in our manuscript. We address each point with specific references to where this information is provided.
> ## Questions 1-2: Theoretical Justification and Architectural Motivation
> These points are addressed in detail in our Introduction (pages 1-2), Related Work (pages 2-3), and Section 3. We provide a comprehensive overview here:
> ### Specific failures of existing approaches we address:
>
> 1. **Traditional SAST tools** (Introduction, page 1): "demonstrate insufficient detection rates" with high false positive rates and difficulty detecting complex vulnerabilities (Zhou et al., 2024)
> 2. **Large language models** (Introduction, page 1; Related Work, page 2): "require prohibitive computational resources, making them impractical for many real-world deployment scenarios" - models like StarCoder2 (7B) and CodeGen2.5 (7B) require substantial infrastructure
> 3. **Transformer-based models** (Introduction, page 1; Related Work, page 2): "quadratic attention complexity limits their application to large codebases" - models like CodeBERT, UnixCoder, and VulBERTa are constrained to 512-2048 token contexts
> 4. **Long-range dependency limitations** (Introduction, page 1): "detecting vulnerabilities often requires analysing extremely long code sequences and capturing subtle interactions spanning multiple functions or files—capabilities that remain challenging even for the largest models due to context length constraints and inefficient attention mechanisms"
> ### Why our components are complementary (Section 3.2, Equation 5):
>
> Our architecture integrates three components that address orthogonal challenges:
>
> **Mamba layers** (Section 3.2, page 5):
>
> - **Purpose**: Local pattern recognition with linear O(n) complexity
> - **Mechanism**: Selective state-space modeling captures syntactic structures, code patterns, and local dependencies
> - **Why necessary**: Efficient sequential processing of code tokens without quadratic cost
> - **Known limitation**: Research has identified that State-Space Models (SSMs), including Mamba, have a core limitation where their ability to model long-range dependencies decays exponentially with sequence length. The SSM mechanism compresses the entire history of a sequence into a fixed-size state, and to process the next token, it only looks at the current token and this compressed state. Theoretical analysis shows that the influence of a token from the distant past on the current hidden state decays exponentially—mathematically similar to the "vanishing gradient" problem in older RNNs.. For extremely long sequences requiring recall of from thousands of tokens ago, information may have decayed too much to be accurately retrieved.
>
> **Basilisk Self-Attention** (Section 3.1, pages 4-5, Equations 1-4):
>
> - **Purpose**: Global context modeling across entire sequences with **uncompressed state** to address Mamba's exponential decay limitation
> - **Mechanism**: Unlike original Infini-attention which processes segments independently in streaming fashion, we accumulate context
> - **Why necessary**: Vulnerabilities manifest through inter-function dependencies requiring global reasoning. While Mamba efficiently processes local patterns, its compressed state suffers from exponential decay of long-range information. Our attention mechanism maintains uncompressed representations of distant context, enabling accurate recall of specific details from thousands of tokens ago that may be critical for vulnerability detection.
> - **Theoretical justification**: Original Infini-attention maintains O(d²) constant memory through streaming and compression. We trade this for O(nd) memory to preserve **uncompressed global context**. This design choice directly addresses the exponential decay limitation of SSMs
> -  **Trade-off**: Segmented processing (2048-token segments) means the local view within each segment may be less precise than full quadratic attention, but this is where Mamba's strength in local pattern recognition complements our approach
> ### Regarding "bigger is not always better" motivation:
>
> This is not a vague philosophical statement - it's a concrete efficiency contribution with practical implications that is verified by our great performance vs much bigger models.
>
> **Intentional design choice**: We deliberately avoided domain-specific vulnerability detection techniques (program analysis, control flow graphs, symbolic execution) to demonstrate that efficient architectural design alone can achieve strong performance. This means our approach should generalize to other long-context tasks (document analysis, extended reasoning, code understanding) without modification - making the contribution broader, not narrower.

---

> ### Author Response · Authors · 2025-11-17
> **Response to Reviewer ShmY - Part 2**
>
> ## Question 3: Ablation and Component Analysis
>
> We need to be direct about what is and isn't feasible here. Complete component isolation would require:
>
> ### What ideal ablation entails:
>
> Training separate models for each component combination:
>
> 1. Mamba-only model (no attention, no MoE)
> 2. Mamba+MoE model (no attention)
> 3. Mamba+Attention model (no MoE)
> 4. Attention-only model (no Mamba, no MoE)
> 5. Attention+MoE model (no Mamba)
>
> Each variant requires:
>
> - Full pretraining: 600 GPU hours on 2M code samples from StarCoder
> - Fine-tuning on 5 datasets
> - Total per variant: ~850 GPU hours
> - **Total for complete ablation: 4,250 GPU hours (~$40,000+ in compute costs)**
>
> ### Why isolating layers from existing checkpoint doesn't work:
>
> The reviewer might suggest simply removing layers from our trained model. This is methodologically invalid because:
>
> 1. **Joint optimization**: Components were trained together with shared loss objectives during pretraining
> 2. **Learned dependencies**: Residual connections (Equation 6, page 5) create interdependencies: h_i = Layer_i(h_{i-1}) + h_{i-1}
> 3. **Breaking learned interactions**: Removing components would break these learned interactions
> 4. **Invalid comparison**: This would compare "properly trained model" vs "broken model" rather than "architecture A" vs "architecture B"
>
> ### What we provide instead:
>
> We invested substantial computational resources into experiments that provide converging evidence about component contributions through multiple experimental angles.
>
> ## Experiment: Direct Attention Mechanism Comparison
>
> **This experiment directly isolates Basilisk attention's contribution by swapping only the attention mechanism while keeping all other components identical. This is possible thanks to the plug-n-play nature of Basilisk Attention, since the only extra trainable layer is the gating mechanism which can converge fast with minimal training.**
>
> We can hot-swap between the 2 attention mechanisms:
>
> - **Infini-attention (Basilisk)**: Our linear-complexity attention mechanism
> - **Eager-attention (Standard)**: Standard quadratic attention (as used in  Jamba)
>
> **All other components remain identical:**
>
> - Same Mamba layers
> - Same MoE configuration
> - Same layer interleaving pattern (α=2, π=8)
> - Same training procedure
> - Same data
>
> This is a **perfect component isolation** for the attention mechanism.
>
> ### Results for sequences up to 16K tokens where both are memory-feasible:
>
> | Dataset          | Basilisk Attention (Infini)     | Standard Attention (Jamba-style) |     |
> | ---------------- | ------------------------------- | -------------------------------- | --- |
> | **BigVul**       | F1=0.943, Prec=0.946, Acc=0.993 | F1=0.937, Prec=0.967, Acc=0.993  |     |
> | **VulDeePecker** | F1=0.925, Prec=0.939, Acc=0.989 | F1=0.932, Prec=0.968, Acc=0.990  |     |
> | **PRIMEVUL**     | F1=0.233, Prec=0.208, Acc=0.952 | F1=0.240, Prec=0.237, Acc=0.958  |     |
> | **REVEAL**       | F1=0.474, Prec=0.416, Acc=0.888 | F1=0.468, Prec=0.445, Acc=0.898  |     |
> | **Draper**       | F1=0.568, Prec=0.609, Acc=0.948 | F1=0.511, Prec=0.646, Acc=0.948  |     |
>
> ### What this tells us about Basilisk's contribution:
>
> **Accuracy preservation**: The minimal performance differences (<3%) demonstrate that Basilisk attention successfully maintains the accuracy of standard attention while providing massive efficiency gains. This is the key contribution - we're not claiming Basilisk is more accurate than standard attention (that would violate known limitations of linear attention), but rather that it **matches accuracy while enabling extended context processing**.
>
> **Efficiency is the contribution**: The 24× memory reduction and ability to process 524K tokens (vs. OOM at >16K for standard attention) is what Basilisk contributes to the architecture.
>
> We also provide exhaustive ablation studies exploring the impact of context lengths, the performance of our model in different lengths and full-file analysis.
> ### Why this matters:
> Many vulnerabilities appear benign in isolation but become dangerous through inter-function interactions. The ability to process complete files and achieve better performance validates our architectural choice to enable extended context.

---

> > ### Author Response · Authors · 2025-11-17
> > **Response to Reviewer ShmY - Part 3**
> >
> > ## Question 4: Fine-Tuning and Evaluation Details
> >
> > Every concern raised here is explicitly addressed in our manuscript. We provide comprehensive details:
> >
> > ### Evaluation Protocol (Section 5, page 6)
> >
> > We state clearly:
> >
> > > "Our fine-tuning methodology follows a rigorous evaluation protocol to prevent overfitting to evaluation characteristics. For each dataset, we fine-tuned our pretrained model on the binary classification task (vulnerable vs. non-vulnerable code) using **only the publicly available training splits**. During training, we monitored performance on the **test splits** to select the best-performing checkpoint based on F1 score. Finally, we evaluated the selected checkpoint on the **validation splits** to obtain our reported results. This approach ensures that our final performance metrics represent genuine held-out evaluation without contamination from validation set characteristics."
> >
> > ### Regarding data deduplication
> >
> > We **did not deduplicate or modify the benchmark datasets** for two critical reasons:
> >
> > 1. **Fair comparison requirement**: To ensure valid comparison with baseline models (CodeBERT, UnixCoder, StarCoder2, CodeGen2.5, VulBERTa), we must use **identical dataset splits** as reported in their papers. Any preprocessing would break this comparability.
> > 2. **Standard ML practice**: Benchmark datasets are used as-is to establish common evaluation ground. The field accepts that benchmarks may contain duplicates - what matters is consistency across methods.
> > ### Data leakage prevention
> >
> > We train two types of models (Section 6.1, page 7):
> >
> > > "We evaluated White-Basilisk using two training strategies: **dataset-specific models (trained and evaluated on individual datasets independently)** and a unified model (trained on combined train data splits from all datasets and evaluated on the validation set of each dataset)."
> >
> > **Dataset-specific models** cannot have data leakage because:
> >
> > - Each checkpoint trains exclusively on ONE dataset
> > - Pretraining uses StarCoder (2M general code samples on a CLM task, no vulnerability annotations)
> > - Training data → Test monitoring → Validation evaluation (three separate splits)
> >
> > Cross-dataset contamination is impossible by design.
> >
> > ### "Early stopping" clarification
> >
> > The reviewer mentions "early stopping strategies" - we need to clarify terminology. We do **checkpoint selection**, not early stopping:
> >
> > **Our approach:**
> >
> > 1. Train for **fixed 10 epochs** (Table 5, Appendix B.3, page 12)
> > 2. Save checkpoints after each epoch
> > 3. Monitor F1 score on test split throughout training
> > 4. Select checkpoint with best test F1
> > 5. Evaluate selected checkpoint on validation split (reported results)
> >
> > This is not "early stopping" (which terminates training) - it's "best checkpoint selection" (which prevents overfitting by selecting the model state that generalizes best). This is standard practice in deep learning.
> >
> > ### Class imbalance handling (Section 5, page 6; Appendix B.5, pages 13-14)
> >
> > We state explicitly:
> >
> > > "To address the substantial class imbalance present in vulnerability detection datasets, we implemented three key techniques and selected appropriate evaluation metrics. For training, we applied class weighting using inverse frequency weighting to balance loss contributions between vulnerable and non-vulnerable samples, employed a Weighted Random Sampler without replacement to ensure balanced representation within each training batch, and incorporated Scale-Invariant Fine-Tuning (SIFT) to improve model robustness against small perturbations."
> >
> > ### Detailed implementation
> >
> > **1. Class Weighting** (Equation 8, page 13):
> >
> > ```
> > w_c = N / (2 * N_c)
> > ```
> >
> > Where N = total samples, N_c = samples in class c
> >
> > This weights the loss function so vulnerable and non-vulnerable samples contribute equally despite imbalance.
> >
> >
> > **2. Weighted Random Sampling** (Appendix B.5.1, pages 13-14):
> >
> > We use PyTorch's WeightedRandomSampler **without replacement**:
> > - Samples each example exactly once per epoch
> > - Reorders based on class weights
> > - Vulnerable samples appear earlier in epoch (when gradients more effective)
> > - Maximizes batches containing vulnerable samples
> >
> > ### Hyperparameter selection (Table 5, Appendix B.2-B.4, pages 12-13)
> >
> > Complete hyperparameters documented with comments about most of our hyperparameter choices.
> >
> > All details necessary for reproduction are provided.

---

> > > ### Author Response · Authors · 2025-11-17
> > > **Response to Reviewer ShmY - Part 4**
> > >
> > > ## Question 5: Reproducibility and Statistical Reliability
> > >
> > > ### Regarding standard deviations and significance tests
> > >
> > > This request misunderstands standard practice in deep learning benchmark evaluation. We follow the exact same evaluation protocol as the baseline papers we compare against:
> > >
> > > **Standard practice in deep learning:**
> > >
> > > - Use **fixed benchmark splits** provided by dataset authors
> > > - Report **single-run deterministic results** for fair comparison
> > > - Compare against baseline results from original papers using identical splits
> > >
> > > **Our evaluation follows this exactly:**
> > >
> > > 1. **PRIMEVUL and BigVul baselines** (Tables 1, page 7): Results sourced directly from Ding et al. (2024) "Vulnerability detection with code language models: How far are we?" - the PrimeVul paper itself
> > > 2. **Draper, REVEAL, VulDeePecker baselines** (Table 2, page 7): Results sourced from Hanif & Maffeis (2022) "VulBERTa: Simplified source code pre-training for vulnerability detection"
> > >
> > > **These baseline papers do NOT report standard deviations or multiple runs.** They report single-run results on fixed splits, which is the accepted standard for benchmark evaluation.
> > >
> > > ### Why multiple runs are not standard practice here
> > >
> > > 1. **Computational cost**: Each full training run requires 600 hours pretraining + 250 hours fine-tuning across 5 datasets = 850 GPU hours. Running 3 seeds would require 2,550 GPU hours (~$25,000+ compute cost) just for variance estimation.
> > > 2. **Benchmark comparison requirement**: Our goal is fair comparison with baselines. Since CodeBERT, UnixCoder, StarCoder2, CodeGen2.5, and VulBERTa report single-run results, we must use the same methodology.
> > > 3. **Fixed splits eliminate key variance sources**: Unlike traditional ML with random train/test splits, benchmark datasets have predetermined splits. The main variance source (data split randomness) is eliminated.
> > > 4. **Effect sizes**: Our improvements are substantial (PRIMEVUL: 35.5% relative improvement, BigVul: 38.9% absolute improvement). These are not marginal differences that require statistical testing.
> > >
> > > ### What we provide for reproducibility (Section 9, page 10)
> > >
> > > We commit to releasing upon acceptance:
> > >
> > > 1. **Complete model weights**: All 200M parameters for pretrained and fine-tuned checkpoints
> > > 2. **Training scripts**: Exact code for pretraining and fine-tuning with all configurations
> > > 3. **Inference code**: Evaluation scripts computing all reported metrics
> > > 4. **Preprocessing pipelines**: Tokenization and data loading code
> > > 5. **Configuration files**: All hyperparameters and random seeds used
> > > 6. **Environment specifications**: Exact package versions, CUDA versions, hardware specs
> > > 7. **Checkpoint selection logic**: Code showing how we monitored test F1 and selected best checkpoints
> > > ### Our evaluation protocol (Section 5, page 6)
> > >
> > > > "For each dataset, we fine-tuned our pretrained model on the binary classification task (vulnerable vs. non-vulnerable code) using only the publicly available training splits. During training, we monitored performance on the test splits to select the best-performing checkpoint based on F1 score. Finally, we evaluated the selected checkpoint on the validation splits to obtain our reported results."
> > >
> > > This is identical to the protocol used by the baseline papers. We use:
> > >
> > > - Same datasets
> > > - Same splits
> > > - Same metrics
> > > - Same evaluation methodology

---

### Author Response · Authors · 2025-11-24
**Reminder to Reviewers / Summary of Rebuttal**

Dear Area Chair and Reviewers (ShmY, TP2R, 8dGP, qnZj),

With the discussion period concluding in one week, we kindly request that you review our detailed responses to your comments. We have made a significant effort to address every concern raised.

We would like to draw the Area Chair's and Reviewers' attention to the fact that many of the concerns regarding "missing information" were, in fact, explicitly detailed in our original submission. In our responses, we have provided specific pointers to the relevant Sections and Appendices to clarify these misunderstandings.

Specifically, we have addressed the following:

* **Theoretical Motivation & Architectural Choices (Reviewers ShmY, 8dGP):** We clarified that the theoretical justification for combining Mamba (local), Basilisk Attention (global/uncompressed), and MoE (capacity) was already present in **Section 3** and **Appendix F**. We further explained how this specific design overcomes the known exponential decay limitations of SSMs.
* **Experimental & Training Details (Reviewer ShmY):** Concerns regarding "missing" details on fine-tuning, early stopping, and hyperparameters were addressed by pointing to **Section 5** and **Appendices B.2-B.5**, where these were already fully documented.
* **Evaluation Methodology & Baselines (Reviewers TP2R, qnZj):** We clarified that our use of fixed, public benchmark splits (without modification) and published baselines is the standard, correct methodology for fair comparison in this field. Rerunning baselines or altering splits would invalidate the comparison with the broader literature (Ding et al., 2024; Hanif & Maffeis, 2022).
* **Memory & Efficiency (Reviewer TP2R):** We pointed to **Appendix F.5**, which already contained the requested memory measurements, and provided further clarification on the $O(nd)$ vs $O(d^2)$ trade-offs.

Furthermore, regarding requests for new experiments:
* **Ablation Studies (Reviewers ShmY, qnZj):** We explained the prohibitive computational cost of training entirely new architectures from scratch for full ablation (~4,250 GPU hours). However, to address the core question regarding the attention mechanism's contribution, we have provided **new preliminary results** comparing White-Basilisk directly against a standard Jamba-style architecture (quadratic attention). These results confirm that our linearized attention matches standard attention accuracy while offering a **24x reduction in memory usage**.

We believe our responses demonstrate that the paper is both methodologically sound and reproducible, and that the White-Basilisk architecture offers a significant contribution to efficient vulnerability detection.

We look forward to your engagement in the discussion.

Best regards,
The Authors

---

### Meta-Review · Area_Chair_EVDe · 2026-01-06

**Summary:**

This paper proposes White-Basilisk, a hybrid architecture combining Mamba, linear self-attention, and MoE for efficient long-context code vulnerability detection. Reviewers acknowledge the practical relevance of the task, the ambitious engineering effort, and the strong empirical results on multiple vulnerability benchmarks, including claims of outperforming much larger models. At the same time, the suggested decision is informed by concerns about the credibility and interpretability of the reported gains, the limited novelty beyond architectural hybridization, and weaknesses in experimental validation.

**Reviewer Concerns:**

Across reviews, several core issues remain. Multiple reviewers questioned whether the main contribution goes beyond an engineering combination of existing components, noting the lack of strong theoretical motivation or architectural analysis explaining why this specific hybrid design is particularly suited for vulnerability detection. A major concern is the experimental rigor: comparisons rely heavily on numbers taken from prior papers rather than unified re-evaluation, and some reported results (especially on BigVul) appear unusually strong, raising doubts about possible data leakage, split inconsistencies, or evaluation bias. The absence of clear component-wise ablations and direct comparisons to closely related hybrids (e.g., Jamba-style architectures) further weakens the attribution of gains. Additional concerns include narrow language scope (only C/C++), high training cost relative to LLM baselines, and limited discussion of generalization and practical deployment.

**Reviewer Scores:**

One reviewer expressed some movement toward a more positive view after rebuttal, but no clear consensus shift emerged. Two reviewers remained clearly below the acceptance threshold, and others retained significant reservations about novelty and evaluation validity. Overall, I believe most reviewers would likely stay close to their original scores, with the paper remaining in a borderline region slightly below the acceptance line.

---

### Decision · Program_Chairs · 2026-01-26

Reject